# Systemic genome-epigenome analysis captures a lineage-specific super-enhancer for *MYB* in gastrointestinal adenocarcinoma

Fuyuan Li[1,5], Shangzi Wang [ID][1,5], Lian Chen [ID][1], Ning Jiang[1], Xingdong Chen [ID][1,2,3,4 ✉] & Jin Li [ID][1 ✉]

## Abstract

**Gastrointestinal adenocarcinoma is a major cancer type for the digestive system, ranking as the top cause of cancer-related deaths worldwide. While there has been extensive research on mutations in protein-coding regions, the knowledge of the landscape of its non-coding regulatory elements is still insufficient. Combining the analysis of active enhancer profiles and genomic structural variation, we discovered and validated a lineage-specific super-enhancer for *MYB* in gastrointestinal adenocarcinoma. This super-enhancer is composed of a predominant enhancer e4 and several additional enhancers, whose transcriptional activity is regulated by the direct binding of HNF4A and MYB itself. Suppression of the super-enhancer downregulated the expression of *MYB*, inhibited downstream Notch signaling and prevented the development of gastrointestinal adenocarcinoma both in vitro and in vivo. Our study uncovers a mechanism driven by non-coding variations that regulate *MYB* expression in a lineage-specific manner, offering new insights into the carcinogenic mechanism and potential therapeutic strategies for gastrointestinal adenocarcinoma.**

**Keyword** Gastrointestinal Adenocarcinoma; MYB; Super-enhancer; Lineage-specific; Epigenetics
**Subject Categories** Cancer; Chromatin, Transcription & Genomics

## Introduction

The overexpression of oncogenes in patients contributes to the development of cancer, with some of these oncogenes becoming therapeutic targets and inspiring the pharmaceutical industry to develop better cancer therapies (Delmore et al, 2011; Vicente-Duenas et al, 2013; Yang et al, 2024). It has been validated that the variations of non-coding regions in the human genome have significant impacts on health (Groschel et al, 2014; Kataoka et al, 2016; Mansour et al, 2014; Northcott et al, 2014; Peifer et al, 2015). For example, a partial loss of the 3′ UTR of the *PD-L1* gene is associated with the upregulation of *PD-L1* expression in various cancer types, including T cell leukemia and B cell lymphoma (Kataoka et al, 2016). Somatic mutations in the *TAL1* enhancer can introduce de novo MYB binding motifs, leading to the formation of a super-enhancer that drives aberrant *TAL1* expression (Mansour et al, 2014). Transcriptional regulation by non-coding regions is highly lineage-specific (Madani Tonekaboni et al, 2019). Thus, further research is needed to better understand the role of non-coding regions in cancer development and progression and to uncover new therapeutic targets for individual cancer type. Gastrointestinal adenocarcinoma is a major cancer type of the digestive system, ranking as the top cause of cancer-related deaths worldwide (Bray et al, 2024). However, knowledge of its lineage-specific variations in non-coding regions remains limited.

Dysregulation of transcriptional programs can be driven by epigenetic alterations targeting non-coding regulatory elements such as enhancers and super-enhancers (SEs) (Leeman-Neill et al, 2023; Zhang and Meyerson, 2020; Zhou and Parsons, 2023). In cancer research, SEs are described as clusters of enhancers in close proximity, marked by H3K27ac, transcription factors or mediator complex to drive the high expression of lineage-specific oncogenes (Hnisz et al, 2013). It has been demonstrated that the recurrent alterations of SEs play an important role in tumorigenesis and malignant phenotypes by interacting with the corresponding promoters via the three-dimensional (3D) structure of genome (Liu et al, 2021; Xing et al, 2019; Zhang et al, 2018; Zhang et al, 2016). For example, SE amplifications increase the expression of oncogenes including *MYC*, *SOX2*, *USP12*, *KLF5*, *ZFP36L2*, thereby promoting the progression of various cancers, including endometrioma and gastrointestinal tumors (Liu et al, 2021; Xing et al, 2019; Zhang et al, 2018; Zhang et al, 2016). Considering the diversity of different types of cancer, significant efforts are demanded to understand the transcriptional regulation of oncogenes by the SEs in gastrointestinal adenocarcinoma, especially in colorectal cancer (CRC) and gastric cancer.

*MYB* is a protein-coding gene, which encodes a protein with a highly conserved DNA-binding domain (Ciciro and Sala, 2021). As

---

[1]State Key Laboratory of Genetics and Development of Complex Phenotype, School of Life Sciences, Human Phenome Institute, Fudan University, Shanghai 200438, China. [2]Fudan University Taizhou Institute of Health Sciences, Taizhou, China. [3]Yiwu Research Institute of Fudan University, Yiwu, Zhejiang, China. [4]National Clinical Research Center for Aging and Medicine, Huashan Hospital, Fudan University, Shanghai 200040, China. [5]These authors contributed equally: Fuyuan Li, Shangzi Wang. ✉E-mail: xingdongchen@fudan.edu.cn; li_jin_lifescience@fudan.edu.cn

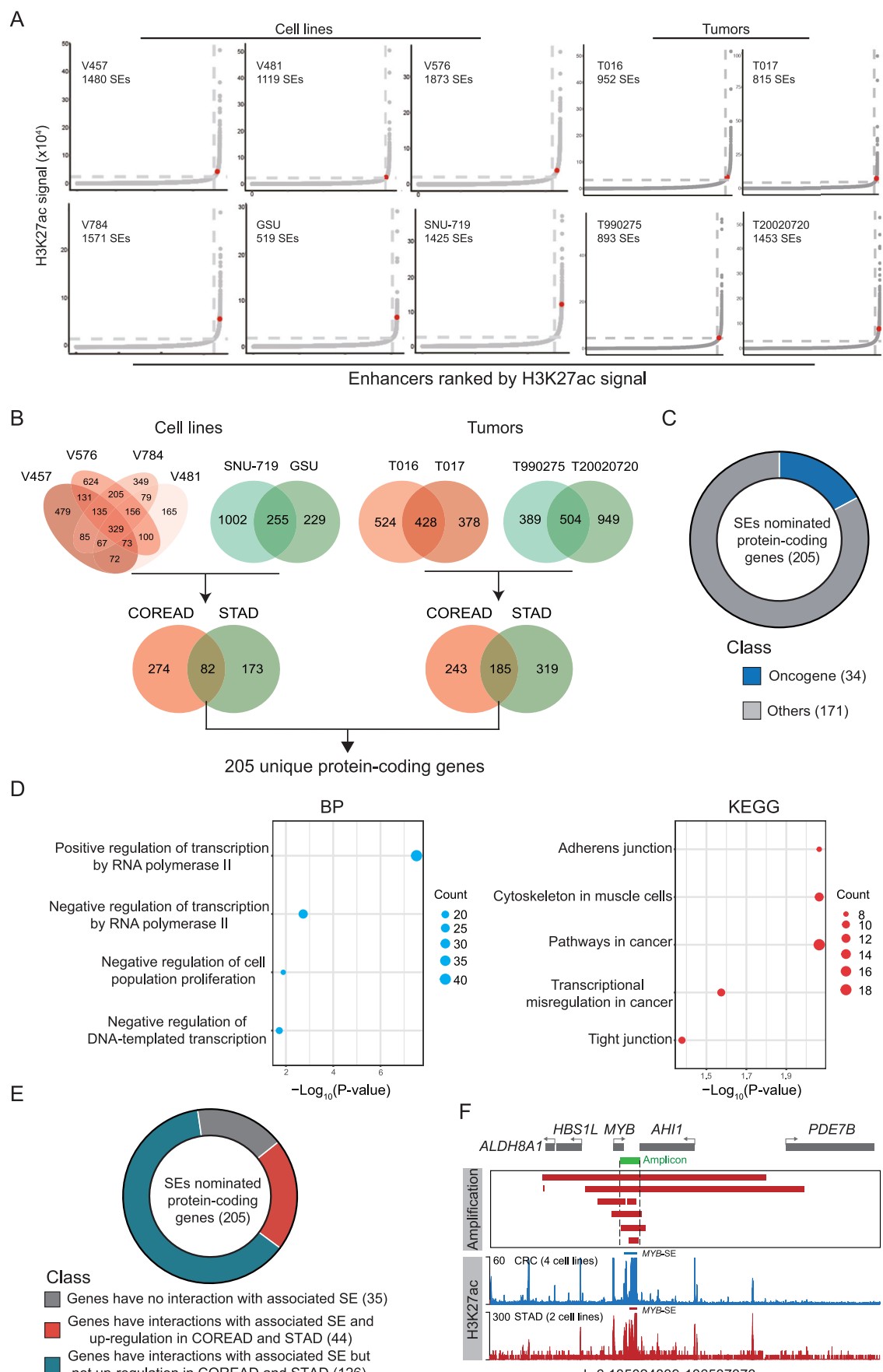

**Figure 1.   A super-enhancer near MYB locus is duplicated in gastrointestinal adenocarcinoma.**

(A) Hockey stick plots for super-enhancer profiles of COREAD and STAD in cancer cell lines. The ranking was calculated by the H3K27ac ChIP-seq signal enrichment. The super-enhancers associated with *MYB* were highlighted in red. (B) Venn diagram showing the shared super-enhancers of various gastrointestinal cancer cell lines or tumor samples, respectively. (C) Pie chart showing oncogene classification of 205 SEs nominated protein-coding genes. (D) The enrichment and functional analysis of 205 SEs nominated protein-coding genes. BP: biological process. KEGG: Kyoto Encyclopedia of Genes and Genomes. The *P*-value was determined by Benjamini correction method. (E) Pie chart showing categorization of 205 SEs nominated protein-coding genes based on enhancer-promoter interaction and expression. (F) The *MYB* amplicons across gastrointestinal adenocarcinoma samples from PCAWG project and merged enhancer profile from cell lines at the *MYB*-associated locus in gastrointestinal adenocarcinoma.

a transcription factor, it regulates the expression of several oncogenes, including *MYC* (Quintana et al, 2011), *TAL1* (Mansour et al, 2014) and *IGF1R* (Andersson et al, 2017). The overexpression of *MYB* is widely observed in cancer cells and it's oncogenic activity has been confirmed in leukemia (Pattabiraman and Gonda, 2013), breast cancer (Quintana et al, 2011), prostate cancer (Acharya et al, 2023), and other cancers (Ramsay and Gonda, 2008). Most known mechanisms underlying *MYB* overexpression involve gene duplication (Lahortiga et al, 2007), translocation (Belloni et al, 2011; Clappier et al, 2007) or intragenic mutations (Hugo et al, 2006). However, the regulation of *MYB* expression by non-coding variation in human genome has not been fully investigated in gastrointestinal adenocarcinoma.

In this study, we identified a lineage-specific SE in gastrointestinal adenocarcinoma that robustly enhances the expression of *MYB* and controls the development of tumor. Within this SE, a predominant enhancer e4 was discovered. The activity of e4 is regulated by the direct binding of HNF4A and MYB and it has a joint effect with other enhancers within this SE. Suppressing the activity of e4 significantly reduced *MYB* expression, inhibited the Notch signaling, decreased the cell viability of CRC and gastric cancer cells and prevented the growth of tumor xenograft. Our study revealed a non-coding variation-based mechanism to affect *MYB* expression in a lineage-specific manner, providing inspiring insights into the carcinogenic mechanism and potential therapeutic strategies for gastrointestinal adenocarcinoma.

## Results

### A lineage-specific super-enhancer near the *MYB* gene is duplicated in gastrointestinal adenocarcinoma

SEs are acquired by cancer cells at key oncogenes (Hnisz et al, 2013). We investigated the landscape of active enhancers in primary tumors and cell lines of colon and rectum adenocarcinoma (COREAD) and stomach adenocarcinoma (STAD) by analyzing the data of chromatin immunoprecipitation-sequencing (ChIP-seq) based on H3K27ac, a marker for active promoters and enhancers, from various datasets deposited in Gene Expression Omnibus (GEO) (Cohen et al, 2017; Okabe et al, 2020; Ooi et al, 2020; Orouji et al, 2022; Zhang et al, 2018). We identified 3433 unique SEs in cancer cell lines and 2137 unique SEs in tumor samples, respectively (Fig. 1A). Considering the shared lineage between colon and stomach, we ought to focus on the SEs common to both COREAD and STAD. Cell lines and tumors shared 82 and 185 SEs, respectively, which are associated with 205 unique protein-coding genes (Fig. 1B and Table EV1), including 34 oncogenes (Fig. 1C). Gene Ontology (GO) (Gene Ontology, 2015) and Kyoto

Encyclopedia of Genes and Genomes (KEGG) (Kanehisa and Goto, 2000) pathway analyses revealed that these genes were predominantly involved in transcriptional regulation and cancer-related process (Fig. 1D; Appendix Fig. S1A). Furthermore, Reactome pathway analysis showed that these genes were involved in canonical cancer-related signaling pathways, such as Notch and TGF-β signaling (Appendix Fig. S1B). These findings suggested that the common SEs may play an important role in the development of gastrointestinal adenocarcinoma.

With their adjacent SEs, 44 genes had physical interactions with SEs based on H3K27ac HiChIP analysis in HT55 and SNU-719 cells (Table EV2) (Liu et al, 2020; Maestri et al, 2023) and were upregulated in COREAD and STAD based on RNA-seq analysis (Fig. 1E; Appendix Fig. S1C). Among them, 11 genes were previously annotated as oncogenes that have physical interactions with their SEs and were upregulated in COREAD and STAD (Fig. EV1A). As a common event of genomic structural variation in cancer cells, enhancer duplication often drives the abnormally high expression of oncogenes (Zhang and Meyerson, 2020). Among these 11 SEs, only the SEs associated with *KLF5* and *MYB* were located within the genomic regions showing recurrent duplication, as observed in a high-resolution whole-genome sequencing dataset from the Pan-Cancer Atlas of Whole Genomes (PCAWG) Project (Consortium, 2020). The *KLF5*-SE duplication had been reported previously (Zhang et al, 2018).

*MYB* is an oncogene that is widely expressed in acute myeloid leukemia (AML) (Ciciro and Sala, 2021; Ramsay and Gonda, 2008). Especially, *MYB* is a gene with an SE present in both tumor samples and cell lines of gastrointestinal adenocarcinoma (Fig. EV1B). The SE (*MYB*-SE) within the amplicon near *MYB* is shown in Figs. 1F and EV1C. Although duplications containing *MYB*-SE were found in many cancer types, *MYB*-SE duplication occurs more frequently in gastrointestinal adenocarcinoma (Fig. EV1D). The information of duplication was derived from WGS data of tumor samples, for which the H3K27ac ChIP-seq data is not available. The information on H3K27ac was derived from cell lines, which have no corresponding WGS data. Therefore, we cannot determine whether the H3K27ac signal of *MYB*-SE is associated with its amplification.

It has been reported that the activity of SEs is specific to cell type and tumor type (Hnisz et al, 2013; Whyte et al, 2013). Therefore, we examined the lineage specificity of the candidate *MYB*-SE in other cancer types. Although the genomic region of the candidate *MYB*-SE was also duplicated in esophageal adenocarcinoma (ESAD), a cancer type closely related in lineage but with remarkable biological differences compared to COREAD and STAD, the ChIP-seq analysis revealed that it did not constitute a SE (Fig. EV1E). In addition, though *MYB* is considered as an oncogene in AML with a potential super-enhancer (Pelish et al, 2015), the candidate *MYB*-

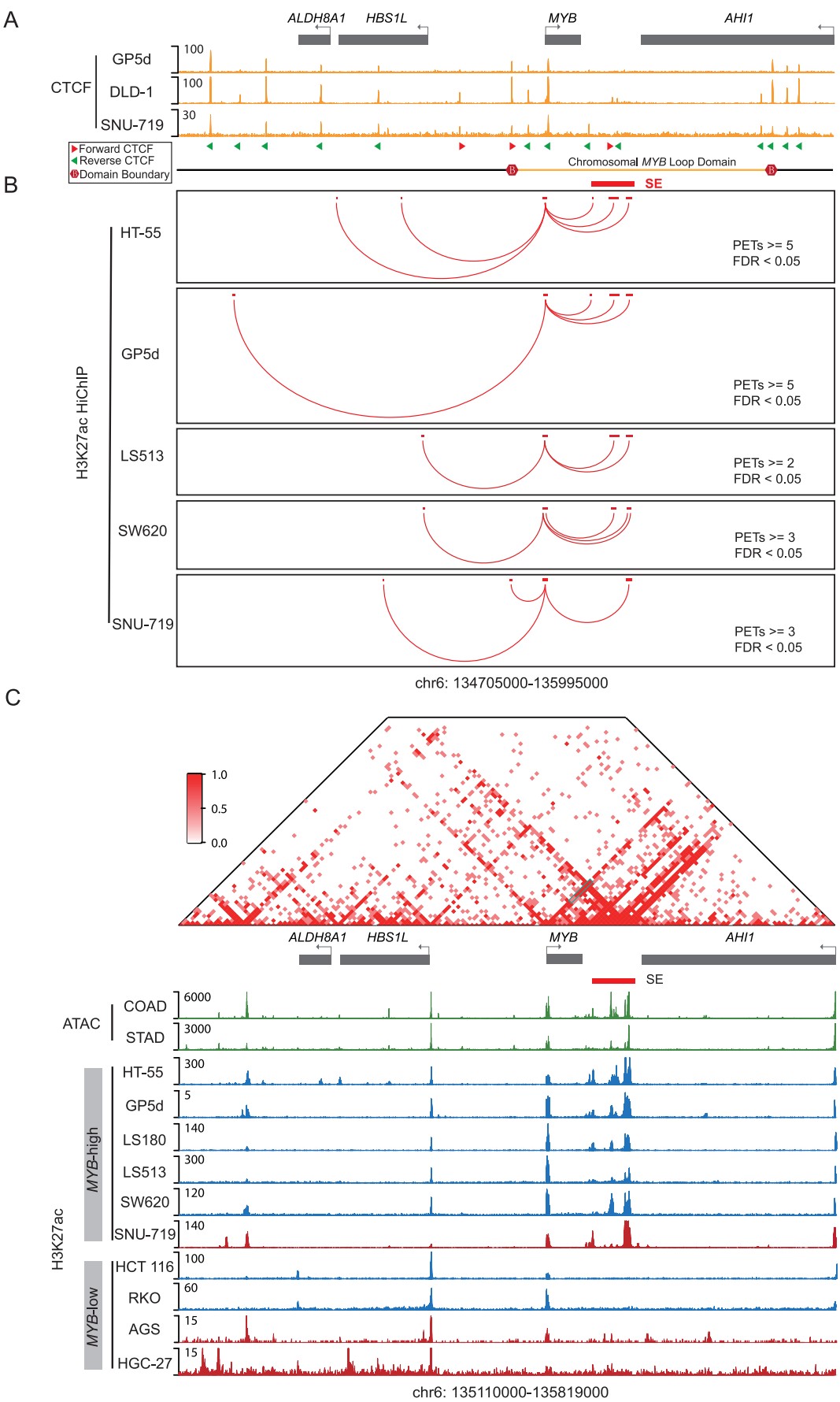

**Figure 2.  Long-range interaction between the super-enhancer and promoter of *MYB* in gastrointestinal adenocarcinoma.**

(A) The CTCF ChIP-seq tracks of multiple gastrointestinal cancer cell lines were presented. The red and green arrows represented the orientation of the CTCF motifs. A schematic of the chromosomal *MYB* loop domain is shown at the bottom. (B) The chromatin loops based on H3K27ac HiChIP data connected the *MYB* promoter to multiple distal elements with high-confidence PETs in various gastrointestinal cancer cell lines. (C) The demonstration of the epigenetic features for the *MYB* associated locus. Top: The chromatin interaction frequency heatmap in the gastrointestinal cancer cell line HT-55 based on H3K27ac HiChIP data. Turquoise box showed the interactions between the *MYB* promoter and the candidate *MYB*-SE. Middle: The ATAC-seq tracks of COREAD and STAD samples from TCGA. Bottom: The H3K27ac ChIP-seq tracks of 10 gastrointestinal cancer cell lines. Blue for COREAD tracks and red for STAD tracks.

SE was not present in AML and many other cancer types (Fig. EV1E). We also tested the presence of *MYB*-SE in other species using a published dataset from mouse (Li et al, 2023). Compared to a well-known oncogene *Myc*, we observed a low H3K27ac ChIP-seq signal (Fig. EV1F) and low expression of *Myb* in mouse CRC cell lines (Fig. EV1G), suggesting that *MYB*-SE is likely specific to humans. We applied liftOver (Kent et al, 2002) to convert the human *MYB*-SE gene coordinates to mouse coordinates (mm10) and found that the DNA sequence underlying the *MYB*-SE is conserved between human and mouse except the enhancer e6 (Fig. EV1F).

Could the existence of the candidate *MYB*-SE impact the expression of *MYB* in gastrointestinal adenocarcinoma? Based on RNA-seq data of normal samples from the Genotype-Tissue Expression project (GTEx) (Carithers et al, 2015; Consortium, 2015) and tumor samples from the Cancer Genome Atlas (TCGA) (Cancer Genome Atlas Research et al, 2013), the expression of *MYB* in COAD, READ and STAD tumors was higher compared to normal tissue (Fig. EV1H). Similar results were obtained from paired sample expression profiles from TCGA (Fig. EV1I). Using cell line data from CCLE (Barretina et al, 2012), we found that *MYB* expression showed only a weak correlation with the copy number of its coding region (Fig. EV1J), indicating the copy number is not a major driver of *MYB* expression. Thus, we hypothesized that the overexpression of *MYB* in COREAD and STAD was driven by the presence of the candidate *MYB*-SE.

To determine the functional significance of *MYB* in gastrointestinal adenocarcinoma, we analyzed the genome-wide CRISPR screen data released by DepMap (Tsherniak et al, 2017). The CRISPR dependency score was normalized such that, for each cell line, the median effect of a set of known non-essential genes (referred to as "negative controls") is 0, and the median effect of known essential genes (referred to as "positive controls") is −1 (Liu et al, 2020). As a result, a lower score indicates a higher likelihood that a gene is dependent in a given cell line, so a cutoff of −0.5 corresponds to half the effect size expected from knocking out genes that are essential for each cell line (Liu et al, 2020). Hence, if a cell line exhibit CRISPR dependency score $<-0.5$ for a given gene, the cell line is dependent on that gene. The *MYB* dependency level identified from the data of DepMap was tightly correlated with the results from CRISPR-based genetic screen by the Sanger Institute (Behan et al, 2019) and shRNA-based screen project by the Broad Institute and Novartis (Root et al, 2006), indicating the robustness of the analysis (Fig. EV2A). Importantly, the cancer cell lines with higher expression levels of *MYB* exhibited an even greater dependence on the *MYB* gene for many types of cancer, including COREAD and STAD (Fig. EV2B). The expression and dependency of *MYB* in a subset of COREAD and STAD cancer cell lines are

shown in Fig. EV2C. These results highlighted the significance of *MYB* overexpression in gastrointestinal adenocarcinoma.

## 3D genomics analysis identified *MYB* candidate functional enhancers in gastrointestinal adenocarcinomas

We next sought to explore how the candidate *MYB*-SE regulates the expression of *MYB* in gastrointestinal cancer. It is widely accepted that the three-dimensional architecture of chromatin plays a crucial role in transcriptional regulation (Davidson and Peters, 2021). Therefore, we first investigated the interaction between *MYB* promoter and candidate *MYB*-SE using a published dataset of ChIP-seq based on the CCCTC-binding factor (CTCF) (Boukaba et al, 2022; Maestri et al, 2023; Sahu et al, 2022), which plays an indispensable role in mediating the formation of topologically associating domains (TADs) and chromatin loops (Davidson and Peters, 2021). Interestingly, the *MYB* promoter and the candidate *MYB*-SE resided within the same chromatin loop domain, defined by mutually convergent CTCFs binding sites (Fig. 2A), suggesting that the *MYB* promoter may interact with the candidate *MYB*-SE through chromatin looping.

By analyzing the published H3K27ac HiChIP data from representative cell lines (Table EV2) (Donohue et al, 2022; Johnstone et al, 2020; Karttunen et al, 2023; Liu et al, 2020; Maestri et al, 2023; Mortenson et al, 2024; Orouji et al, 2022), we investigated the physical interactions between the *MYB* promoter and the enhancers within the candidate *MYB*-SE by identifying promoter-enhancer loops. The results of the HiChIP analysis indicated that the constituent enhancers of the candidate *MYB*-SE had highly confident pair-end-tags (false discovery rate (FDR) < 0.05) linking them to the *MYB* promoter in *MYB*-high/-dependent cell lines (Fig. 2B) and a *MYB*-high/-independent HT-29 cell line (Appendix Fig. S2A). In contrast, these pair-end-tags were absent in *MYB*-low/-independent cell lines (including HCT 116, RKO, AGS cell lines) and normal human primary colonic epithelial cells (Appendix Fig. S2B). The independence of HT-29 on *MYB* can be attributed to either potential alternative mechanism of survival or the absence of SE (Appendix Fig. S2A).

We next analyzed the chromatin landscape using the Assay for Transposase-Accessible Chromatin with high throughput sequencing (ATAC-seq) of patient samples (Corces et al, 2018) and H3K27ac ChIP-seq of COREAD and STAD cancer cell lines with high or low *MYB* expression (Fig. 2C). We found that the candidate *MYB*-SE region exhibited a chromatin-accessible state (Fig. 2C), and *MYB*-high cancer cell lines displayed strong H3K27ac signals in this region (Fig. 2C). In contrast, *MYB*-low cancer cell lines showed minimal enrichment of H3K27ac signals in the candidate

*MYB*-SE region (Fig. 2C). The situation became more complex when *MYB* dependency was considered. We identified the presence of the candidate *MYB*-SE in both *MYB*-high/-dependent and some *MYB*-high/-independent cell lines but not in *MYB*-low/-independent cell lines (Appendix Fig. S2C). Taken together, these results revealed a correlation between the presence of the candidate *MYB*-SE and the expression or dependency of *MYB* in gastrointestinal adenocarcinoma.

To identify the predominant enhancer(s) within the *MYB*-SE which may drive *MYB* overexpression, we divided the *MYB*-SE into seven individual enhancers, based on ATAC-seq and H3K27ac ChIP-seq signals in this region (Fig. 3A). The initial analysis was performed using self-transcribing active regulatory region sequencing (STARR-seq) (Sahu et al, 2022), a method to directly identify enhancers in a genome-wide manner. The results revealed that e4 exhibited stronger transcriptional activity compared to the other enhancers of the *MYB*-SE in the *MYB*-high and -dependent GP5d cells (Appendix Fig. S3A). We further validated the significance of e4 in *MYB* expression with the luciferase assay in HT-55 and SNU-719 cells (Fig. 3B). In contrast, the activity of e4 was not detected in HEK293-FT cells (Appendix Fig. S3B), consistent with the lineage specificity of *MYB*-SE in gastrointestinal adenocarcinomas. To mimic enhancer duplication in the gastrointestinal adenocarcinomas, we generated a plasmid containing an additional copy of e4 (Appendix Fig. S3C), which indeed exhibited stronger luciferase activity (Fig. 3C).

In addition, we tested the function of e4 in regulating *MYB* expression by manipulating its transcriptional activity endogenously. We performed an improved CRISPR-based interference (CRISPRi) assay (details are provided in the Methods and Appendix Methods) (Liu et al, 2021; Yeo et al, 2018) to suppress the proposed constituent enhancers in four *MYB*-high gastrointestinal adenocarcinoma cell lines (Fig. 3D). The results revealed that repression of e4 led to a significant reduction in *MYB* expression across all four cell lines, whereas suppression of the other six enhancers had only minimal effects (Appendix Fig. S3D). Moreover, the effects of CRISPRi targeting e4 on *MYB* expression were attenuated in the *MYB*-high/-independent CCK-81 cells (Appendix Fig. S3D). The importance of e4 in *MYB* expression was further validated using two additional sgRNAs in HT-55 and SNU-719 cells (Fig. 3E). In contrast, activation of e4 by CRISPR-based activation assay (CRISPRa, details are provided in the Methods and Appendix Methods) (Sanson et al, 2018) in AGS and HGC-27 cell lines, two *MYB*-low/-independent cell lines, dramatically increased *MYB* expression (Appendix Fig. S3E). These results suggested that e4 within the *MYB*-SE is the predominant enhancer driving *MYB* expression.

It is widely accepted that a single enhancer may regulate the expression of multiple genes within the same genomic region (Fukaya et al, 2016). Based on the analysis of H3K27ac HiChIP data, we found that the HiChIP anchor of e4–7 exhibited the strongest physical interactions with the *MYB* promoter across all five *MYB*-high/-dependent cell lines (Fig. EV3A). However, we also found four additional genes, *HBS1L*, *BCLAF1*, *MAP7* and *IL20RA*, whose promoters interacted with e4–7 in more than one cell line (Fig. EV3A). We therefore investigated whether e4 controls the transcription of genes other than *MYB*. CRISPRi-mediated suppression of e4 had minimal effects on the expression of *HBS1L*, *BCLAF1*, *MAP7*, and *IL20RA* as well as other genes surrounding

*MYB* in two cell lines (Fig. EV3B,C). Although the expression of *AHI1*, a gene located near *MYB*, was modestly affected, no direct interaction with e4 was identified by the analysis of H3K27ac HiChIP data (Fig. EV3D,E). This suggested that e4 does not directly target *AHI1* promoter via a chromatin loop extrusion mechanism. However, other enhancer regulation models, such as phase separation (Hnisz et al, 2017; Tang et al, 2022), could still contribute to the gene's regulation.

The enhancers within a SE often function cooperatively to regulate the expression of a single gene (Hnisz et al, 2017). Intestinally, we also observed interactions among multiple enhancers within the *MYB*-SE (Fig. EV3F,G). The luciferase assay demonstrated that the combination of e4 with other constituent enhancers within the *MYB*-SE exhibited higher activity than e4 alone (Fig. 3F), indicating a joint effect of these enhancers within the *MYB*-SE on transcriptional regulation. Collectively, our findings revealed the predominant role of e4 within *MYB*-SE in controlling *MYB* expression.

## HNF4A controls the activity of e4 for the *MYB*-SE via direct interactions

After confirming the predominant role of e4 within the *MYB*-SE, we next investigated the potential transcription factors (TFs) that bind to e4 and regulate its activity. Based on the analysis of phastCons scores (Siepel et al, 2005) and DNase I hypersensitive sites (Consortium, 2012) on e4, we observed that this region exhibited high DNA sequence conservation and a chromatin-accessible state across numerous species (Fig. 4A), making it feasible for the downstream analysis. To identify TFs that may bind to e4, we performed motif analysis using publicly available ATAC-seq datasets from tumor samples, including 81 primary COREAD samples (from 38 donors) and 41 primary STAD samples (from 21 donors) from TCGA (Corces et al, 2018). By employing the FIMO (Grant et al, 2011) and TRAP (Thomas-Chollier et al, 2011) (see Methods and Appendix Methods for details), we identified that multiple potential transcription factor families may bind to e4 (Figs. 4A and EV4A).

To test whether the expression of *MYB* can be regulated by the binding of specific TFs, we performed genome editing using the CRISPR-Cas9 system to disrupt DNA motifs within e4. We found that disruption of the motifs associated with the HNF4, EBF, ASCL, SOX, or SNAI families led to a significant reduction in *MYB* expression (Fig. EV4B,C). Deletion of specific TF-binding motif within e4 showed that the ASCL, EHF, EBF, HNF4, SNAI, and SOX motifs are important for maintaining e4 activity, as demonstrated by the luciferase assay (Figs. 4B and EV4D). Especially, the depletion of EBF or HNF4 motifs had the most dramatic effects on e4 activity.

Based on the results of motif analysis and genome editing, we focused on two TFs, EBF1 (EBF family motif) and HNF4A (HNF4 family motif), for further investigation. First, we validated the binding of EBF1 and HNF4A to e4 by ChIP-qPCR assay in HT-55 and SNU-719 cells (Fig. 4C). Importantly, knockdown of *HNF4A*, but not *HNF4G*, reduced the expression of *MYB* and potently inhibited the activity of e4 in the luciferase assay (Figs. 4D,E and EV4E,F). In contrast, knockdown of *EBF1* did not affect *MYB* expression (Fig. EV4E,G). Pharmacological inhibition of HNF4A using its antagonist, BI-6015, also suppressed *MYB* expression and e4 activity

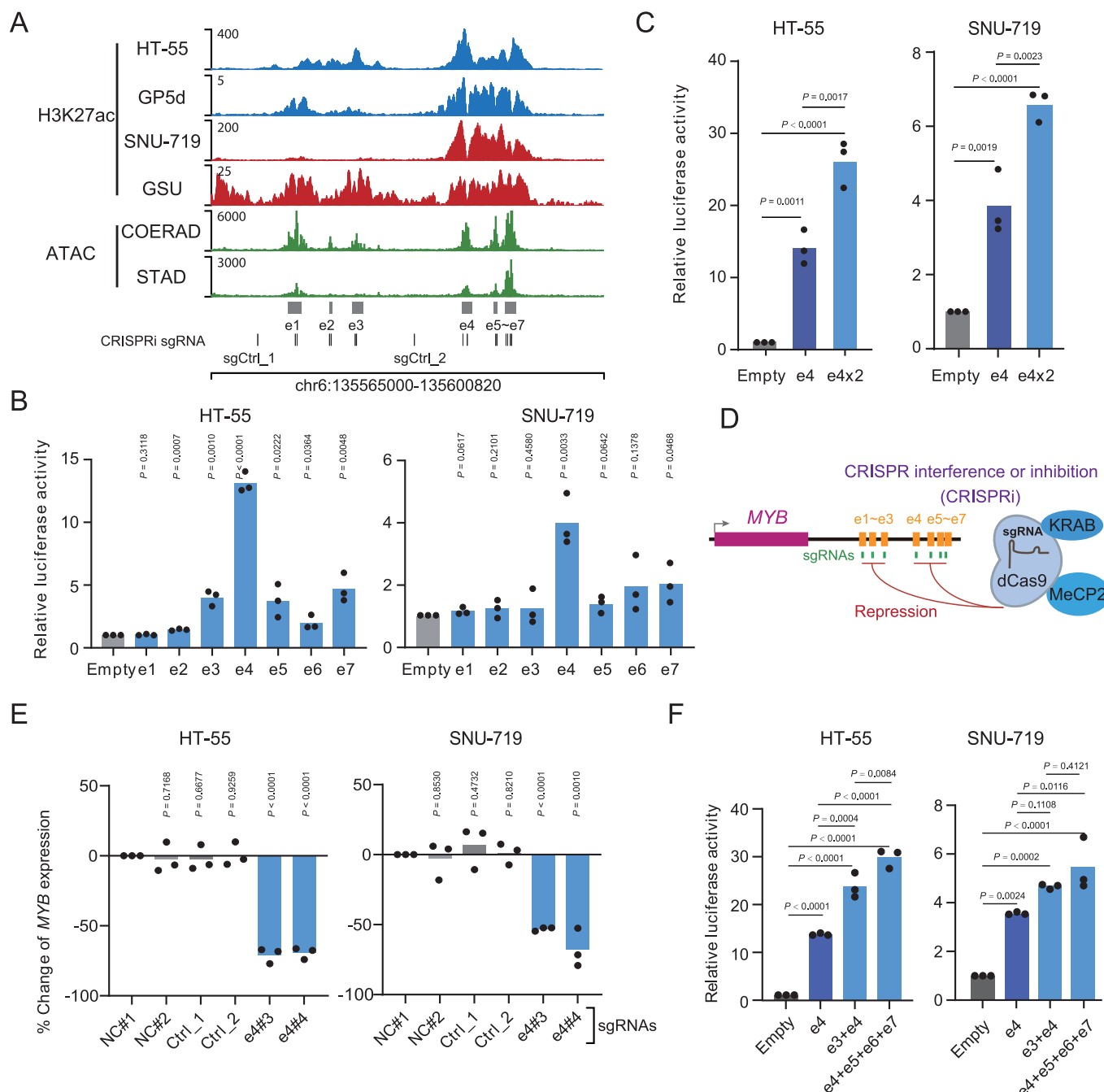

**Figure 3. The identification of the predominant enhancer driving *MYB* expression.**

(A) H3K27ac ChIP-seq signal from gastrointestinal adenocarcinoma cancer cell lines and ATAC-seq signal from tumor samples revealed the constituent enhancers e1–e7 for the *MYB*-SE, sgCtrl_1 and sgCtrl_2: sgRNA targeted negative control region. (B) Luciferase reporter assay measuring the activity of candidate enhancers in HT-55 and SNU-719 cells. The empty pGL3-promoter vector was used as a negative control. $N = 3$. All values are mean ± S.D. The *P*-value was determined by two-sided Student's *t*-test. (C) Luciferase reporter assay measuring the activity of single or duplicated enhancer e4 (2 × e4) in HT-55 and SNU-719 cells. The empty pGL3-promoter vector was used as a negative control. $N = 3$. All values are mean ± S.D. The *P*-value was determined by One-way ANOVA. (D) Schematic diagram of the strategy to validate the predominant enhancer by CRISPRi. (E) The expression of *MYB* upon CRISPRi-mediated repression of e4. Two separate sgRNAs, as sg-e4#3 and sg-e4#4, were used to target e4. NC, non-targeted control. $N = 3$. All values are mean ± S.D. The *P*-value was determined by two-sided Student's *t*-test. (F) Luciferase reporter assay measuring the activity of single or combinational enhancer in HT-55 and SNU-719 cells. The empty pGL3-promoter vector was used as a negative control. $N = 3$. All values are mean ± S.D. The *P*-value was determined by One-way ANOVA. Source data are available online for this figure.

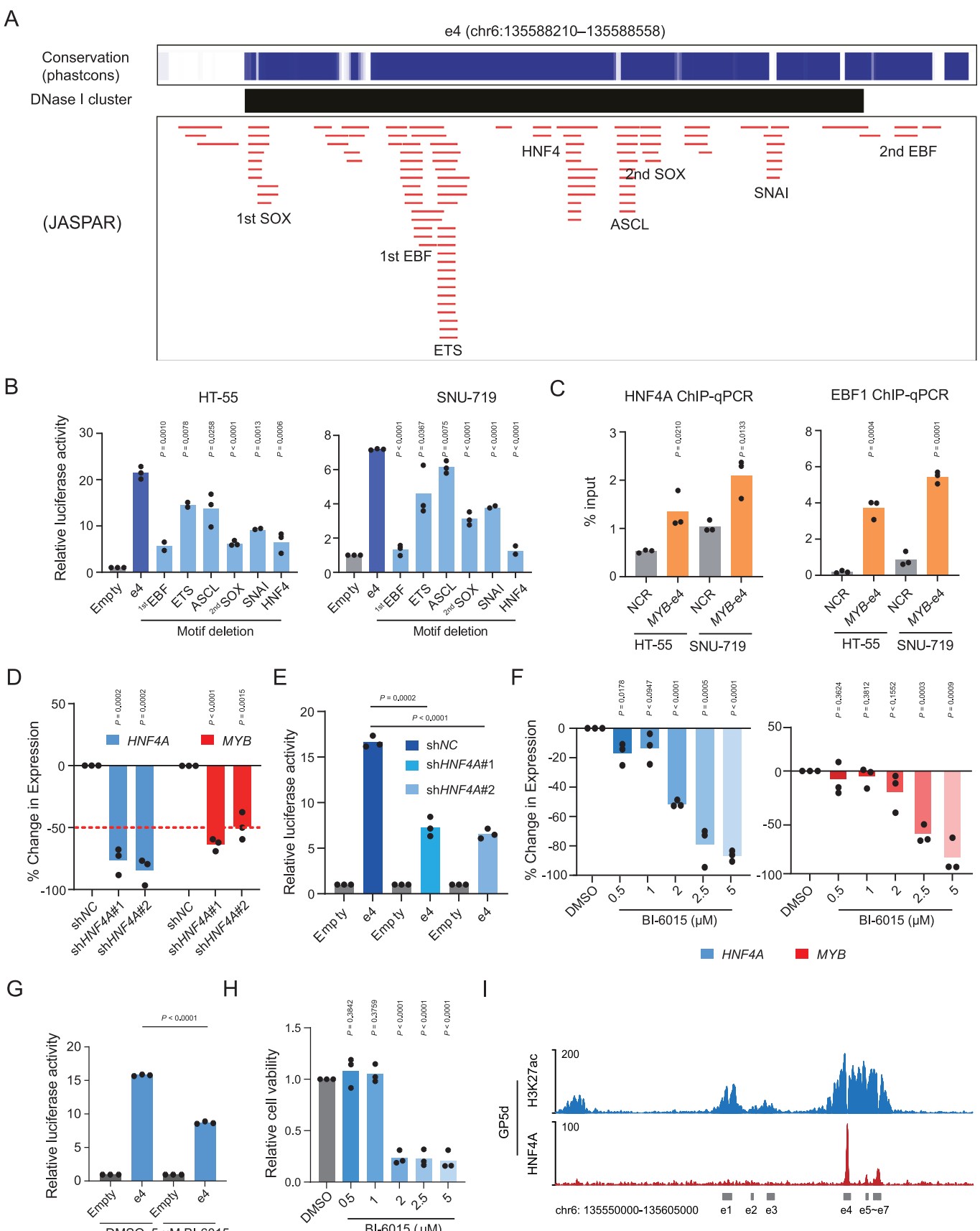

(A) Motif analysis of e4 sequence. Top: phastCons scores (0:1 range) representing the conservation of e4 DNA sequence. Middle: Cluster of DNase I hypersensitive sites demonstrating the accessibility of e4. Bottom: Results of JASPAR-based prediction for transcription factors binding motifs in e4. (B) Luciferase reporter assay measuring the activity of e4 upon the deletion of the binding motifs. $N = 2$–3. All values are mean ± S.D. The *P*-value was determined by two-sided Student's *t*-test. (C) ChIP-qPCR analysis indicating the enrichment of transcription factors HNF4A and EBF1 at e4. NCR: Negative control region. $N = 3$. All values are mean ± S.D. The *P*-value was determined by two-sided Student's *t*-test. (D) The expression of *HNF4A* and *MYB* upon HNF4A knockdown by shRNAs. NC, non-targeted control. $N = 3$. All values are mean ± S.D. The *P*-value was determined by two-sided Student's *t*-test. (E) Luciferase reporter assay measuring the activity of e4 upon HNF4A knockdown in HT-55 cells. $N = 3$. All values are mean ± S.D. The *P*-value was determined by One-way ANOVA. (F) The expression of *HNF4A* and *MYB* upon the treatment of HNF4A antagonist BI-6015 in HT-55 cells. $N = 3$. All values are mean ± S.D. The *P*-value was determined by two-sided Student's *t*-test. (G) Luciferase reporter assay measuring the activity of theenhancer e4 following the treatment of HNF4A antagonist BI-6015 in HT-55 cells. $N = 3$. All values are mean ± S.D. The *P*-value was determined by two-sided Student's *t* test. (H) Cell viability of HT-55 cells after 48 h treatment of BI-6015, measured by Cell Counting Kit-8 (CCK-8). $N = 3$. All values are mean ± S.D. The *P*-value was determined by two-sided Student's *t*-test. (I) ChIP-seq tracks for H3K27ac and HNF4A in the *MYB*-SE associated locus in GP5d cells. Source data are available online for this figure.

(Fig. 4F,G). Interestingly, treatment with BI-6015 also inhibited the growth of HT-55 cells (Fig. 4H). We further validated that HNF4A binds to e4, e5, e6 and e7 in GP5d cells using HNF4A ChIP-seq (Sahu et al, 2022) data. Among these regions, the binding peak at e4 exhibited the highest signal (Fig. 4I). Additionally, we found several candidate functional transcription factor motifs in other enhancers, suggesting that these transcription factors may bind to multiple enhancers to collaboratively regulate the activity of the *MYB*-SE (Fig. EV4H). In brief, these results suggested that HNF4A can bind to e4 and regulate *MYB* expression by modulating the activity of e4 within the *MYB*-SE.

## MYB activates cancer-related genes in gastrointestinal adenocarcinomas

To systematically investigate the regulatory functions of MYB in gastrointestinal adenocarcinoma, we performed ChIP-seq analysis for MYB in HT-55 and SNU-719 cells. The canonical binding motif of MYB was identified by the software HOMER (Heinz et al, 2010) and presented at Fig. 5A. We also identified motifs for several transcription factors (TFs) in MYB-binding DNA sequences, including AP1, KLF5, ETS1, FOXA1 and other cancer-related TFs (Appendix Fig. S4A), suggesting that these TFs may work together with MYB in transcriptional regulation. We observed that MYB mainly bound to the promoter and distal intergenic regions (Fig. 5B,C), suggesting that MYB likely binds to regulatory elements, including enhancers and SEs.

Assigning the MYB binding sites to their nearest genes identified a total of 1240 genes in both HT-55 and SNU-719 cells (Appendix Fig. S4B and Table EV3). The analysis for GO and KEGG term enrichment revealed that these genes were involved in apoptosis, cancer development, and transcription regulation (Fig. 5D). Specifically, we found that MYB bound to e4, e6, and e7 within the *MYB*-SE based on ChIP-seq and ChIP-qPCR analysis (Fig. 5E; Appendix Fig. S4C). Knocking down *MYB* significantly inhibited e4 activity in the luciferase assay (Appendix Fig. S4D), indicating that MYB can regulate the transcriptional activity of e4. Thus, an auto-regulatory loop, a typical feature of lineage-specific oncogenes (Durbin et al, 2018; Jiang et al, 2020; Ott et al, 2018; Saint-Andre et al, 2016), may exist between MYB and *MYB*-SE. In addition, we found that MYB can bind to SEs of other cancer-related genes (Fig. 5F; Appendix Fig. S4E). Specifically, CRISPRi-based suppression of MYB binding sites in HT-55 cells downregulated the expression of *GATA6* and *ZFP36L2*, two well-known genes with oncogenic roles in gastrointestinal adenocarcinoma (Chia et al, 2015; Xing et al, 2019) (Fig. 5G).

To investigate the genes whose expression is regulated by MYB, we performed RNA sequencing (RNA-Seq) analysis on *MYB*-depleted HT-55 cells. Differential expression analysis identified 532 upregulated genes and 496 downregulated genes in the *MYB*-depleted cells comparing to cells with a scrambled control (Appendix Fig. S5A). Integrative analysis of RNA-seq and ChIP-seq data revealed 407 genes as potential direct targets of MYB (Appendix Fig. S5B and Table EV4). Gene set enrichment analysis (GSEA) showed that MYB targeted genes, which were down-regulated upon *MYB* depletion, were related to Notch signaling (Appendix Fig. S5C). These findings indicated that MYB directly controlled the expression of multiple cancer-related genes, including itself, which may contribute to the development of gastrointestinal adenocarcinoma.

## Targeting e4 suppressed the growth of gastrointestinal cancer cells in vitro and in vivo

Considering the significance of MYB in regulating the expression of *MYB* and multiple oncogenes, we hypothesized that *MYB* is essential for the high proliferation rate of gastrointestinal cancer cells. To validate this, we first used an shRNA system to knock down *MYB* in HT-55 and SNU-719 cells. We observed that suppressing the expression of *MYB* significantly inhibited cell growth (Figs. 6A and EV5A,B). To characterize the effects of e4 on cellular function, we suppressed its activity by CRISPRi system and observed a significant decrease of cell growth as well (Figs. 6B and EV5C). Additionally, suppression of e4 prevented colony formation (Fig. 6C) and reduced the positive rate of EdU staining, a marker for cell proliferation, in HT-55 cells (Fig. 6D). Mechanistically, *MYB* suppression downregulated the expression of genes related to Notch signaling (Fig. EV5D). Notably, treatment with the Notch antagonist BMS-986115 also inhibited HT-55 cell growth (Fig. 6E).

We then tested the significance of *MYB* for gastrointestinal adenocarcinoma in vivo using xenograft model with HT-55 cells in nude mice. We observed that both the volume and weight of the tumor xenografts were significantly reduced by suppressing e4 activity or knockdown of the endogenous *MYB* (Figs. 6F–H and EV5E–G). The expression of Ki-67, a marker of cell proliferation, was decreased following e4 inhibition or *MYB* knockdown in the xenografts (Figs. 6I–J and EV5H). Taking these results collectively, we concluded that *MYB* functions as a potential oncogene in promoting the development of gastrointestinal adenocarcinomas.

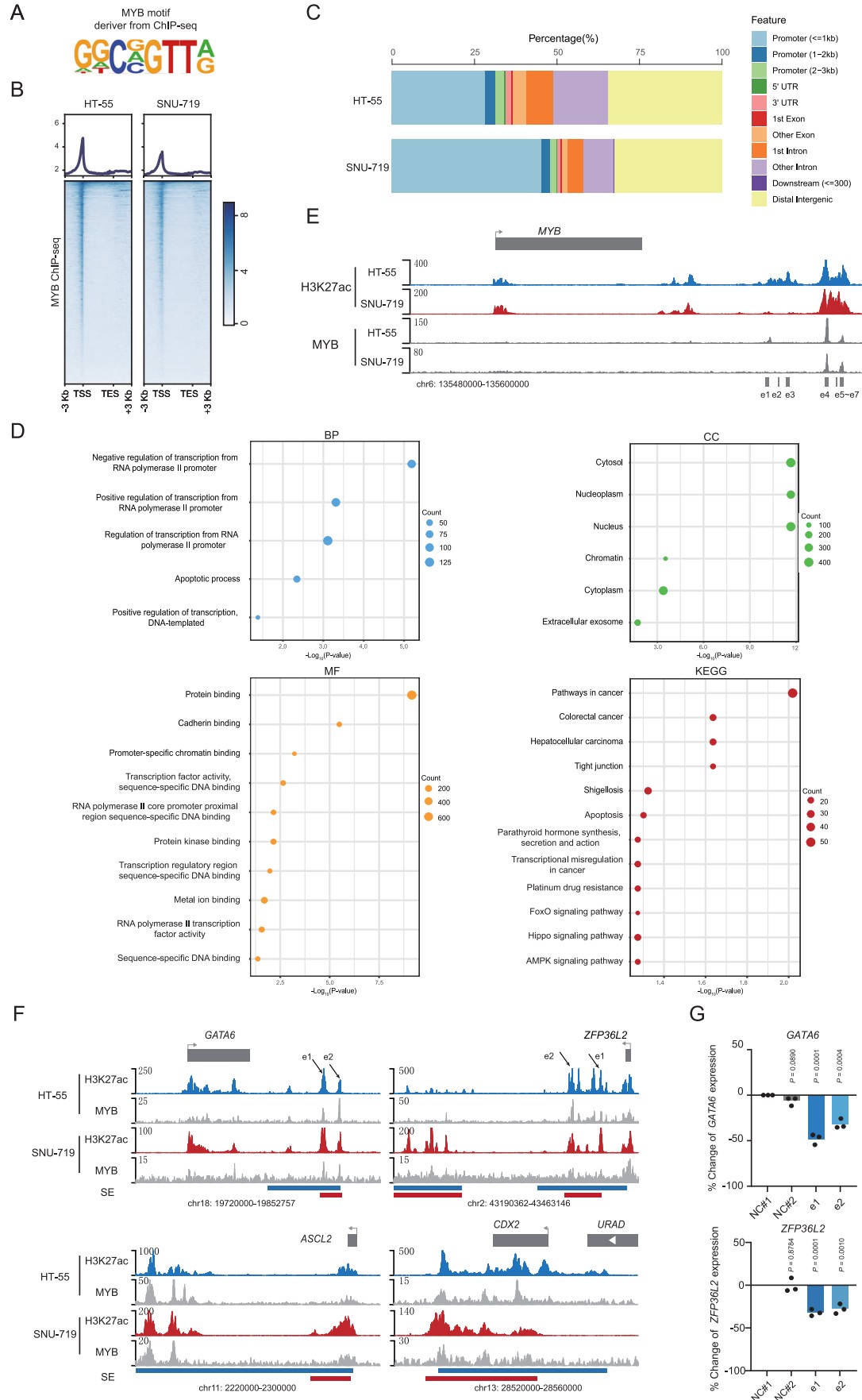

**Figure 5. MYB contributes to the high expression of cancer-related genes in gastrointestinal adenocarcinoma.**

(A) Prediction of MYB's DNA-binding motifs based on MYB ChIP-seq data. (B) Heatmaps showing the distribution of MYB binding sites cross the whole genome in HT-55 and SNU-719 cells. Genes in the rows were sorted in decreasing order of signal intensity. TSS, transcription start site. TES, transcription end site. (C) Bar charts illustrating the types of MYB binding sites in HT-55 and SNU-719 cells. (D) Functional enrichment analysis of the genes nearest to the MYB binding sites. The *P*-value was determined by Benjamini correction method. (E) ChIP-seq tracks of MYB in HT-55 and SNU-719 cells showing MYB bound to its super-enhancer. (F) MYB bound to the candidate super-enhancers of various cancer-related genes. (G) Validation of the functions of the candidate super-enhancers for *GATA6* and *ZFP36L2* by CRISPRi. *N* = 3. All values are mean ± S.D. The *P*-value was determined by two-sided Student's *t*-test. Source data are available online for this figure.

# Discussion

Deciphering the biological and pathological basis of gastrointestinal adenocarcinoma could improve its diagnosis and therapy. By systematically investigating the active enhancer profiles of tumor samples as well as cell lines, we identified a lineage-specific super-enhancer for *MYB*, which contains a dominant enhancer e4 in gastrointestinal adenocarcinoma. Through an auto-regulatory loop, MYB regulates gastrointestinal adenocarcinoma development by directly controlling the transcription of multiple carcinogenic genes related to Notch signaling. These results suggested the specific role of *MYB* in the carcinogenesis of gastrointestinal adenocarcinoma (Fig. 6K).

Various types of genomic alterations at the *MYB* locus have been detected in human cancer, such as duplication (Lahortiga et al, 2007), translocation (Belloni et al, 2011; Clappier et al, 2007), gene-fusion (Belloni et al, 2011), and intragenic mutations (Hugo et al, 2006). However, these genomic alterations can only account for *MYB* expression in a subset of cancers. Increasing evidence suggests that the abnormal activation of enhancers represents a major epigenetic aberration in various types of cancer (Liu et al, 2021; Morton et al, 2019; Terekhanova et al, 2023; Zhang et al, 2018; Zhang et al, 2016). In the present study, we performed a comprehensive analysis of a super-enhancer (SE) that forms a loop with the *MYB* promoter and specifically regulates its expression in gastrointestinal adenocarcinoma. Especially, the predominant enhancer e4 interacts with other enhancers within *MYB*-SE to regulate its transcriptional activity. These results confirmed the existence of "facilitators", as the elements contributing to the classical enhancer function without inherent activities (Blayney et al, 2023). It is possible that some of low-activity enhancers act as facilitators for e4 within the *MYB*-SE. We also found *MYB*-SE is frequently duplicated in gastrointestinal adenocarcinoma in comparison to other cancer types. Although the duplication for genomic region of *MYB*-SE has been reported in some of the T-ALL samples (Lahortiga et al, 2007; O'Neil et al, 2007), the *MYB*-SE region in ALL is not functional for *MYB* expression, as the enhancers in ALL for *MYB* are located 116 kb away from the *MYB*-SE (Li et al, 2021; Stadhouders et al, 2014).

The MYB protein is a critical therapeutic target for various cancer types, such as T cell acute lymphoblastic leukemia (T-ALL) (Pattabiraman and Gonda, 2013), breast cancer (Li et al, 2016), and prostate cancer (Acharya et al, 2023). In this study, we expanded the understanding of MYB by uncovering its essential role in gastrointestinal cancer. Our findings indicated that MYB binds to e4 within the *MYB*-SE to regulate its expression, thereby forming a self-regulatory circuit characteristic of lineage-specific master transcription factors (Durbin et al, 2018; Jiang et al, 2020; Ott

et al, 2018; Saint-Andre et al, 2016). We also found that HNF4A regulated the transcription of *MYB* as well as the growth of gastrointestinal adenocarcinoma via binding to e4 within the *MYB*-SE. Given that an mRNA-based strategy has been developed to target HNF4A in preclinical models for liver fibrosis (Yang et al, 2021), similar strategy can be used to control the development of gastrointestinal adenocarcinoma with HNF4A-dependent over-expression of *MYB*.

In addition to its functions in cancer progression, *MYB* is also involved in the developmental process of neural tissues (Malaterre et al, 2008), colonic crypts (Malaterre et al, 2007), breast tissue (Li et al, 2016) as well as hematopoietic system (Jiang et al, 2006). We showed that the suppression of e4 within the *MYB*-SE, which downregulated the expression of *MYB*, was able to reduce the growth of gastrointestinal adenocarcinoma in vitro and in vivo. This finding highlighted the significance of e4 in *MYB* function and may inspire further studies about the *MYB* associated non-coding variants in the development of non-malignant tissues at the population level.

In summary, our study identified a *MYB* associated lineage-specific super-enhancer, containing a predominant enhancer e4 and a few potential facilitators, which regulated the development of gastrointestinal adenocarcinoma. These findings underscore the potential of non-coding elements, such as super-enhancers, as viable therapeutic targets for cancer treatment.

# Methods

**Reagents and tools table**

| Reagent/Resource | Reference or Source | Identifier or Catalog Number |
|---|---|---|
| **Experimental models** | | |
| HEK 293-FT cells (*H. sapiens*) | Cobioer | CBP60438 |
| HT-55 cells (*H. sapiens*) | Cobioer | CBP60012 |
| SNU-719 cells (*H. sapiens*) | Cobioer | CBP60511 |
| GP5d cells (*H. sapiens*) | Sigma | 95090715 |
| CCK-81 cells (*H. sapiens*) | Cobioer | CBP60581 |
| AGS cells (*H. sapiens*) | Cell Bank of Chinese Academy of Sciences | SCSP-5262 |
| HGC-27 cells (*H. sapiens*) | Cell Bank of Chinese Academy of Sciences | TCHu 22 |

| Reagent/Resource | Reference or Source | Identifier or Catalog Number |
|---|---|---|
| BALB/c-Nude (*M. musculus*) | GemPharmatech | D000521 |
| **Recombinant DNA** | | |
| pLKO.1-puro | Biofeng | 8453 |
| Lenti-dCas9-KRAB-MeCP2-Blast | Liu et al (2021) | N/A |
| Lenti-dCas9-VP64-Blast | Addgene | 61425 |
| Lenti-Cas9-Blast | Addgene | 52962 |
| LentiGuide-puro | Addgene | 52963 |
| pXPR-502 | Addgene | 96923 |
| pGL3-Promoter | Addgene | 212939 |
| pRL *Renilla* | Promega | E2271 |
| pGL3-*MYB*-e1 | This study | N/A |
| pGL3-*MYB*-e2 | This study | N/A |
| pGL3-*MYB*-e3 | This study | N/A |
| pGL3-*MYB*-e4 | This study | N/A |
| pGL3-*MYB*-e5 | This study | N/A |
| pGL3-*MYB*-e6 | This study | N/A |
| pGL3-*MYB*-e7 | This study | N/A |
| pGL3-*MYB*-e4×2 | This study | N/A |
| pGL3-*MYB*-e3 + e4 | This study | N/A |
| pGL3-*MYB*-e4 + e5 + e6 + e7 | This study | N/A |
| Additional plasmids | This study | Appendix Table S1 |
| **Antibodies** | | |
| Rabbit anti-c-MYB | Abcam | ab45150 |
| Rabbit anti-c-MYB | Proteintech | 17800-1-AP |
| Rabbit anti-HNF4α | Abcam | ab181604 |
| Rabbit anti-EBF1 | Millipore | AB10523 |
| Mouse anti-ACTB | CWBio | CW0096M |
| Goat anti-rabbit HRP | ABclonal | AS014 |
| Goat anti-mouse HRP | ABclonal | AS003 |
| Mouse anti-Ki-67 | Cell Signaling Technology | 9449S |
| **Oligonucleotides and sequence-based reagents** | | |
| All primers and more information | This study | Table EV5 |
| All sgRNAs, shRNA and more information | This study | Table EV5 |
| **Chemicals, enzymes, and other reagents** | | |
| DMEM high glucose | Gibco | 11995065 |
| RPMI 1640 | Gibco | 11875119 |
| Fetal bovine serum (FBS) | Gibco | 10099141C |
| Penicillin-Streptomycin | Gibco | 15140122 |
| 0.25% Trypsin-EDTA | Gibco | 25200114 |

| Reagent/Resource | Reference or Source | Identifier or Catalog Number |
|---|---|---|
| Trizol | Takara | 9109 |
| CalPhos Mammalian Transfection Kit | Clontech | 631312 |
| DynaGreen™ Protein A/G | Thermo Fisher Scientific | 80105G |
| FastDigest *Esp3* I | Thermo Fisher Scientific | FD0454 |
| FastAP | Thermo Fisher Scientific | EF0654 |
| Lipofectamine 3000 transfection reagent | Thermo Fisher Scientific | L3000015 |
| Opti-MEM | Gibco | 31985 |
| PEI | Polysciences | 24765 |
| Kpn I-HF® | New England Biolabs | R3142S |
| Mlu I-HF® | New England Biolabs | R3198S |
| Xho I | New England Biolabs | R0146S |
| T4 Polynucleotide Kinase (T4 PNK) | New England Biolabs | M0201L |
| T4 DNA Ligase Reaction Buffer | New England Biolabs | B0202S |
| T7 DNA ligase | New England Biolabs | M0318L |
| Luna® Universal qPCR Master Mix | New England Biolabs | M3003E |
| LunaScript RT SuperMix | New England Biolabs | M3010L |
| Q5® Hot Start High-Fidelity DNA | New England Biolabs | M0493L |
| Q5 Site-Directed Mutagenesis kit | New England Biolabs | E0554S |
| 10 mM dNTPs | New England Biolabs | N0446S |
| NEBuffer™ 2 | New England Biolabs | B7002S |
| *EcoR*I-HF® | New England Biolabs | R3101L |
| *Age*I-HF® | New England Biolabs | R3552L |
| 0.5 M DTT | Beyotime | ST041-10ml |
| Dual Luciferase Reporter Gene Assay Kit | Beyotime | RG027 |
| Cell Counting Kit-8 | Beyotime | C0040 |
| Ampicillin | Beyotime | ST008 |
| Puromycin Dihydrochloride | Beyotime | ST551-250mg |
| Polybrene | Beyotime | C0351-1ml |
| Blasticidin S HCl | Beyotime | ST018-5ml |
| DEPC Water | Beyotime | R0022 |
| DH5α Chemically Competent Cell | Tsingke | TSC-C14 |
| DMOS | Sigma-Aldrich | D2650-100ML |
| Pierce™ DAB Substrate Kit | Thermo Fisher Scientific | 34002 |
| Pierce™ BCA Protein assay Kit | Thermo Fisher Scientific | 23227 |

| Reagent/Resource | Reference or Source | Identifier or Catalog Number |
|---|---|---|
| Cell-Light EdU Apollo488 In Vitro Kit | RIBOBIO | C10310-3 |
| BI-6015 | MedChemExpress | HY-108469 |
| BMS-986115 | MedChemExpress | HY-12860 |
| Matrigel | Corning | 354248 |
| Software | | |
| GraphPad Prism 9 | GraphPad | https://www.graphpad.com/ |
| R | Bioconductor | https://www.r-project.org/ |
| SAMtools | Li et al (2009) | http://samtools.sourceforge.net/ |
| Bowtie2 | Langmead and Salzberg (2012) | https://bowtie-bio.sourceforge.net/index.shtml |
| DEseq2 | Love et al (2014) | https://github.com/thelovelab/DESeq2 |
| HiC-Pro | Servant et al (2015) | https://github.com/nservant/HiC-Pro |
| HOMER | Heinz et al (2010) | http://homer.ucsd.edu/homer/ |
| Liftover | Kent et al (2002) | https://genome.ucsc.edu/cgi-bin/hgLiftOver |
| MACS2 | Zhang et al (2008) | https://github.com/macs3-project/MACS |
| STAR | Dobin et al (2013) | https://github.com/alexdobin/STAR |
| hichipper | Lareau and Aryee (2018) | https://hichipper.readthedocs.io/en/latest/index.html |
| DAVID | Sherman et al (2022) | https://davidbioinformatics.nih.gov/ |
| BEDTools | Quinlan and Hall (2010) | https://bedtools.readthedocs.io/en/latest/index.html |
| ChIPseeker | Yu et al (2015) | https://github.com/YuLab-SMU/ChIPseeker |
| gTrack | Hadi et al (2020) | https://github.com/mskilab-org/gTrack |
| HTSeq | Anders et al (2015) | https://github.com/simon-anders/htseq |

## Cell lines and cell culture

AGS and HGC-27 cells were obtained from the Cell Bank of Chinese Academy of Sciences. HT-55, HEK293-FT, SNU-719, and CCK-81 cells were obtained from Nanjing Cobioer Biosciences company. AGS, HGC-27, and SNU-719 cells were cultured in RPMI 1640 medium (Gibco, #11875119) supplemented with 10% FBS (Gibco, #10099141C) and 1% penicillin-streptomycin (Gibco, #15140122). HEK293-FT, HT-55, and CCK-81 cells were cultured in DMEM high glucose medium (Gibco, #11995065) supplemented with 10% FBS and 1% penicillin-streptomycin. All cells were verified by short tandem repeat analysis and tested negative for

mycoplasma using the Mycoplasma qPCR Detection Kit (Beyotime, #C0303S).

## Luciferase reporter assay

Luciferase reporter assays were performed as previously described (Zhang et al, 2016). Firstly, e4 was amplified from the HT-55 cell's genomic DNA, then e4 regions were cloned upstream of the pGL3-promoter vector (Addgene, #212939) using XhoI (NEB, #R0146S) and MluI (NEB, #R3198S) restriction enzyme sites. For cloning the $2 \times e4$, e4 region was cloned using XhoI and MluI restriction enzyme sites, followed the another e4 region was cloned using KpnI (NEB, #R3142S) and MluI restriction enzyme sites. For the combination of e4 and other constituent enhancers, other enhancers were cloned using XhoI and MluI restriction enzyme sites into 2×e4 vector. The enhancer luciferase reporter constructs were co-transfected with a control luciferase construct, named *Renilla* (Promega, #E2271), into cells using PEI (Polysciences, #24765) or lipofectamine 3000 (Invitrogen, #L3000015). The luciferase signal was first normalized to the *Renilla* luciferase signal and then normalized to the signal from the empty pGL3-promoter vector. Primers used for cloning are listed in Table EV5.

## CRISPR-mediated DNA motif cutting, enhancer repression and activation

According to the ATAC-seq peaks within the candidate *MYB*-SE, the DNA sequence of the enhancers was obtained. Then, CRISPR/Cas9 sgRNAs were designed using the CRISPick tool from the Broad Institute (Doench et al, 2016; Sanson et al, 2018) and negative control, non-targeting sgRNAs were used as previously reported (Liu et al, 2021). For DNA motif cutting, cells were first infected with lenti-Cas9-blast vector (Addgene, #52962) and selected with 10 μg/ml of blasticidin (Beyotime, #ST018-5ml) for at least 7 days. Cells stably expressing Cas9 then subsequently infected with LentiGuide-Pour vector (Addgene, #52963) carrying either non-targeting sgRNAs or sgRNAs targeting the DNA motif and selected with 2 μg/ml puromycin (Beyotime, #ST551-250mg) for at least 5 days before any molecular or cellular assays. For enhancer repression, cells were first infected with lenti-dCas9-KRAB-MeCP2-blast vector (Liu et al, 2021) and selected with 10 μg/ml of blasticidin for at least 7 days. Cells stably expressing dCas9-KRAB-MeCP2 then subsequently infected with LentiGuide-Pour vector carrying either non-targeting sgRNAs or sgRNAs targeting the *MYB* enhancers and selected with 2 μg/ml puromycin for at least 5 days before any molecular or cellular assays. For enhancer activation, cells were first infected with lenti-dCas9-VP64-blast vector (Addgene, #61425) and selected with 10 μg/ml of blasticidin for at least 7 days. Cells stably expressing dCas9-VP64 then subsequently infected with pXPR-502 vector (Addgene, #96923) carrying either negative control sgRNAs or sgRNAs targeting e4 and selected with 2 μg/ml puromycin for at least 5 days before any molecular or cellular assays. All sgRNA sequences are listed in Table EV5.

## Cell growth assays and colony formation assays

For cell growth assays, selected cells were collected. 5000 cells were seeded into 96-well plates (100 μL/well). After 5 days, 10 μL of Cell Counting Kit-8 (CCK-8) (Beyotime, #C0040) was added into each well according to the manufacturer's instructions and incubated for

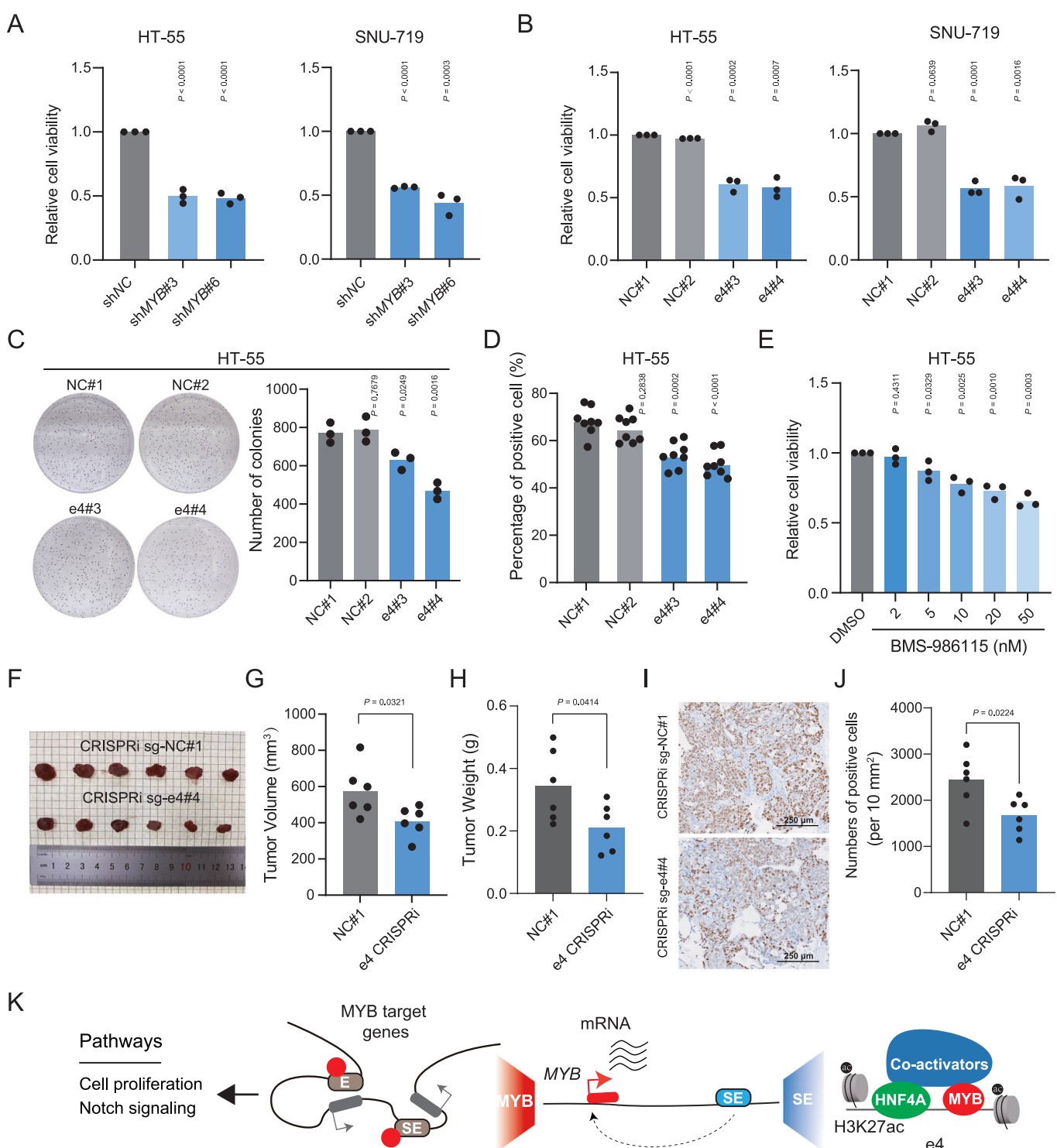

1 h. Optical density was measured at 450 nm using a microplate reader. For colony formation assays, 5000 cells were seeded into 6-well plates (2 mL/well). The cell culture medium was changed every three days. After culture for 18 days, cells were fixed with methanol (100%) and stained with crystal violet solution (25% methanol and 75% crystal violet) for 20 min at room temperature.

**In vivo xenograft tumor assays**

All animal experiments were performed in accordance with procedures approved by the Fudan University ethical committee (IDM2024011). All the mice were maintained under a 12 h light/12 h dark cycle at constant temperature (23 °C) and specific pathogen-free conditions with free

**Figure 6.   The predominant enhancer of *MYB* as a master regulator in the development of gastrointestinal adenocarcinoma.**

(A, B) Cell viability of HT-55 and SNU-719 cells upon *MYB* knockdown (A) or repression of e4 (B), measured by Cell Counting Kit-8 (CCK-8). N = 3. All values are mean ± S.D. The *P*-value was determined by two-sided Student's *t*-test. (C) The effects of repression of e4 on colony formation of HT-55 cells. N = 3. All values are mean ± S.D. The *P*-value was determined by two-sided Student's *t*-test. (D) The effects of repression of e4 on cell proliferation of HT-55 cells as measured by EdU staining. N = 8. All values are mean ± S.D. The *P*-value was determined by two-sided Student's *t*-test. (E) Cell viability of HT-55 cells upon the treatment of Notch antagonist BMS-986115, measured by Cell Counting Kit-8 (CCK-8). N = 3. All values are mean ± S.D. The *P*-value was determined by two-sided Student's *t*-test. (F–H) The growth of xenografts derived from HT-55 cells was significantly inhibited after repression of e4. N = 6. All values are mean ± S.D. The *P*-value was determined by two-sided Student's *t*-test. (I) Representative pictures of Ki-67 staining in xenografts. Scale bar, 250 μm. (J) The average number of Ki-67 positive cells per 10 mm$^2$ of xenograft tumor tissue. N = 6. All values are mean ± S.D. The *P*-value was determined by two-sided Student's *t*-test. (K) Schematic diagram: HNF4A and MYB bind to e4 within *MYB*-SE to activate the expression of *MYB*. The overexpressed MYB protein directly upregulated its target genes and promoted the development of gastrointestinal adenocarcinoma. Source data are available online for this figure.

access to food and water. A total 1 million HT-55 cells with or without CRISPRi system mediated e4 repression or shRNA system mediated *MYB* knock-down were resuspended in 100 μL serum-free DMEM and supplement with 100 μL Matrigel (Coring, #FAL-354248) (total volume 200 μL). Then the cells were subcutaneously injected into the flanks of female nude mice (BALB/c, Nu/Nu, 5–6 weeks old, female from GemPharmatech). Tumor growth was examined every 4–5 days, and tumor length and width were measured using calipers. Until tumor volume reached approximately 1000–1500 mm$^3$ (the tumor size not exceeded 2 cm in any dimension), xenograft tumor-bearing mice were sacrificed, the tumor xenografts were isolated and weighted. Tumor volume was calculated suing the following formula: (length × width$^2$) × 0.52. Animals that developed infections or severe side effects during the course of the experiment were intentionally excluded to maintain the integrity of the data and to avoid confounding factors that could impact the interpretation of the results.

### Public data usage

This study used TCGA publicly available ATAC-seq data were downloaded from NCI Genomic Data Commons data portal (UTR: https://gdc.cancer.gov/about-data/publications/ATACseq-AWG), publicly PCAWG SV data were downloaded from PCAWG data portal (UTR: https://dcc.icgc.org/releases/PCAWG/consensus_sv), the remaining publicly available data were downloaded from GEO. Accession numbers of public dataset used in this study listed in Table EV2.

### Quantification and statistical analysis

All statistical comparisons between two groups were performed by GraphPad Prism software 9.0 using a two-tailed unpaired *t*-test, unless specified. The variance between the statistically compared groups was similar.

## Data availability

The datasets produced in this study are available in the following databases: MYB ChIP-seq data: Gene Expression Omnibus GSE234202. RNA-seq data: Gene Expression Omnibus (GEO) GSE234203.

The source data of this paper are collected in the following database record: biostudies:S-SCDT-10_1038-S44320-025-00098-1.

## Peer review information

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

## Acknowledgements

This work was supported by MOST 2020YFA0803600, 2018YFA0801300, NSFC 32071138 and SKLGE-2118 to JL, MOST 2019YFC1315804, 2019FY101103 and NSFC 82073637, 82122060 to XC. We thank Yueqiang Song for technical support and useful discussions.

## Author contributions

**Fuyuan Li**: Conceptualization; Data curation; Formal analysis; Validation; Investigation; Visualization; Methodology; Writing—original draft; Writing—review and editing. **Shangzi Wang**: Conceptualization; Data curation; Formal analysis; Validation; Investigation; Visualization; Methodology; Writing—original draft; Writing—review and editing. **Lian Chen**: Conceptualization; Data curation; Formal analysis; Validation; Investigation; Visualization; Methodology; Writing—original draft; Writing—review and editing. **Ning Jiang**: Conceptualization; Data curation; Formal analysis; Validation; Investigation; Visualization; Methodology; Writing—original draft; Writing—review and editing. **Xingdong Chen**: Conceptualization; Data curation; Formal analysis; Supervision; Funding acquisition; Validation; Investigation; Visualization; Methodology; Writing—original draft; Writing—review and editing. **Jin Li**: Conceptualization; Data curation; Formal analysis; Supervision; Funding acquisition; Validation; Investigation; Visualization; Methodology; Writing—original draft; Writing—review and editing.

Source data underlying figure panels in this paper may have individual authorship assigned. Where available, figure panel/source data authorship is listed in the following database record: biostudies:S-SCDT-10_1038-S44320-025-00098-1.

## Disclosure and competing interests statement

The authors declare no competing interests.

# Expanded View Figures

**Figure EV1.   The gastrointestinal specificity of *MYB*-SE.**

(A) Venn diagram showing the overlap between oncogenes associated with SEs and upregulated genes with physical interactions with SEs in gastrointestinal adenocarcinoma. (B) Venn diagram showing the shared SEs and their associated genes between gastrointestinal cancer cell lines and tumor samples, with oncogenes highlighted in red. (C) ChIP-seq signal tracks showing the enrichment of H3K27ac at the *MYB*-SE locus. The blue denoted COREAD tracks and red denoted STAD tracks; The cell line information provides a detailed description of the cell lines depicted in Fig. 1F. (D) Focal duplications harboring *MYB*-SE, shorter than 1 Mb in length, were selected for further analysis across 37 cancer types in the PCAWG project. (E) H3K27ac ChIP-seq tracks at the *MYB* locus in cell lines representing each cancer type. ESAD: esophageal adenocarcinoma; BLCA: bladder cancer; PADD: pancreatic adenocarcinoma; LIHC: liver hepatocellular carcinoma; UCEC: uterine corpus endometrial carcinoma; LUAD: lung adenocarcinoma; BRCA: breast cancer; PRAD: prostate adenocarcinoma; LAML: acute myeloid leukemia. (F) ChIP-seq signal tracks of H3K27ac at the *Myb* and *Myc* loci in mouse CRC cell lines. The region of *MYB*-SE is conserved between humans and mice except the enhancer e6. However, the activity of human gastrointestinal adenocarcinoma cancer-specific super-enhancer of *MYB* (converted to mm10 genome) is repressed in this region in mouse CRC cell lines. (G) Expression levels of *Myb* and *Myc* in mouse CRC cell lines, based on RNA-seq data. (H) The expression of *MYB* in cancer tissues and normal tissues, from the TCGA and GTEx databases, was analyzed using Gene Expression Profiling Interactive Analysis 2 (GEPIA2) (Tang et al, 2019). The exact *P*-value were not provided in GEPIA2. The *P*-value was determined by two-sided Student's *t*-test. ***$P \leq 0.001$; Box plots represent the distribution of MYB expression across different groups. The center line within each box denotes the median. The lower and upper bounds of the box indicate the first quartile (Q1, 25th percentile) and third quartile (Q3, 75th percentile), respectively. The whiskers extend to the minimum and maximum values within 1.5× interquartile range (IQR) from Q1 and Q3. Individual points represent paired data and are connected by lines, while potential outliers beyond 1.5× IQR are displayed as separate dots. (I) The expression of *MYB* in COREAD or STAD samples and their corresponding pair-matched normal tissues. The value for the y-axis is $\log_2$ transformed (count+1). $N = 32$ for COREAD or STAD. NS: no significance. Box plots represent the distribution of *MYB* expression across different groups. The center line within each box denotes the median. The lower and upper bounds of the box indicate the first quartile (Q1, 25th percentile) and third quartile (Q3, 75th percentile), respectively. The whiskers extend to the minimum and maximum values within 1.5× interquartile range (IQR) from Q1 and Q3. Individual points represent paired data and are connected by lines, while potential outliers beyond 1.5× IQR are displayed as separate dots. (J) Correlation between *MYB* expression and *MYB* copy number, calculated using Spearman's rank correlation coefficient based on the data from CCLE.

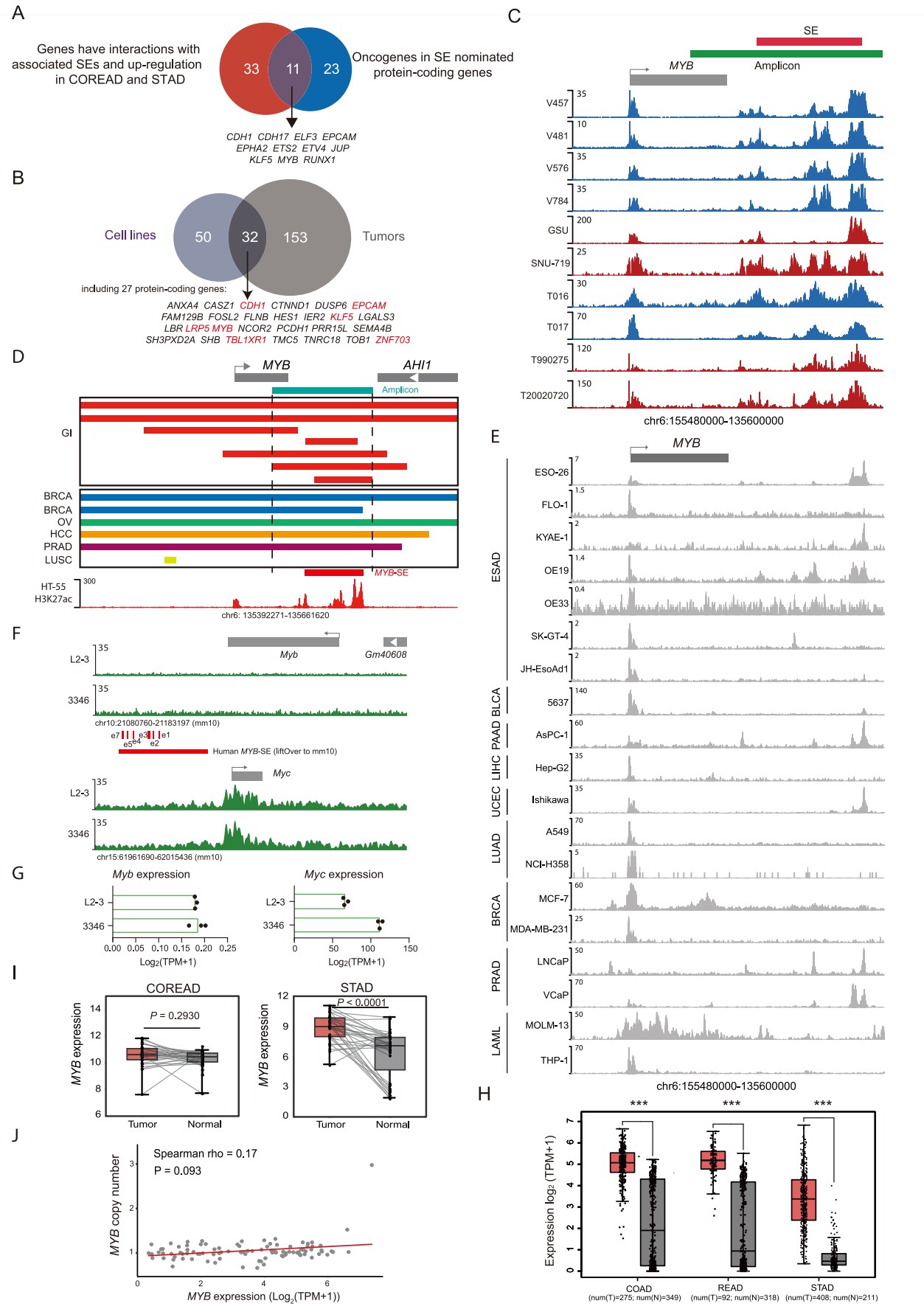

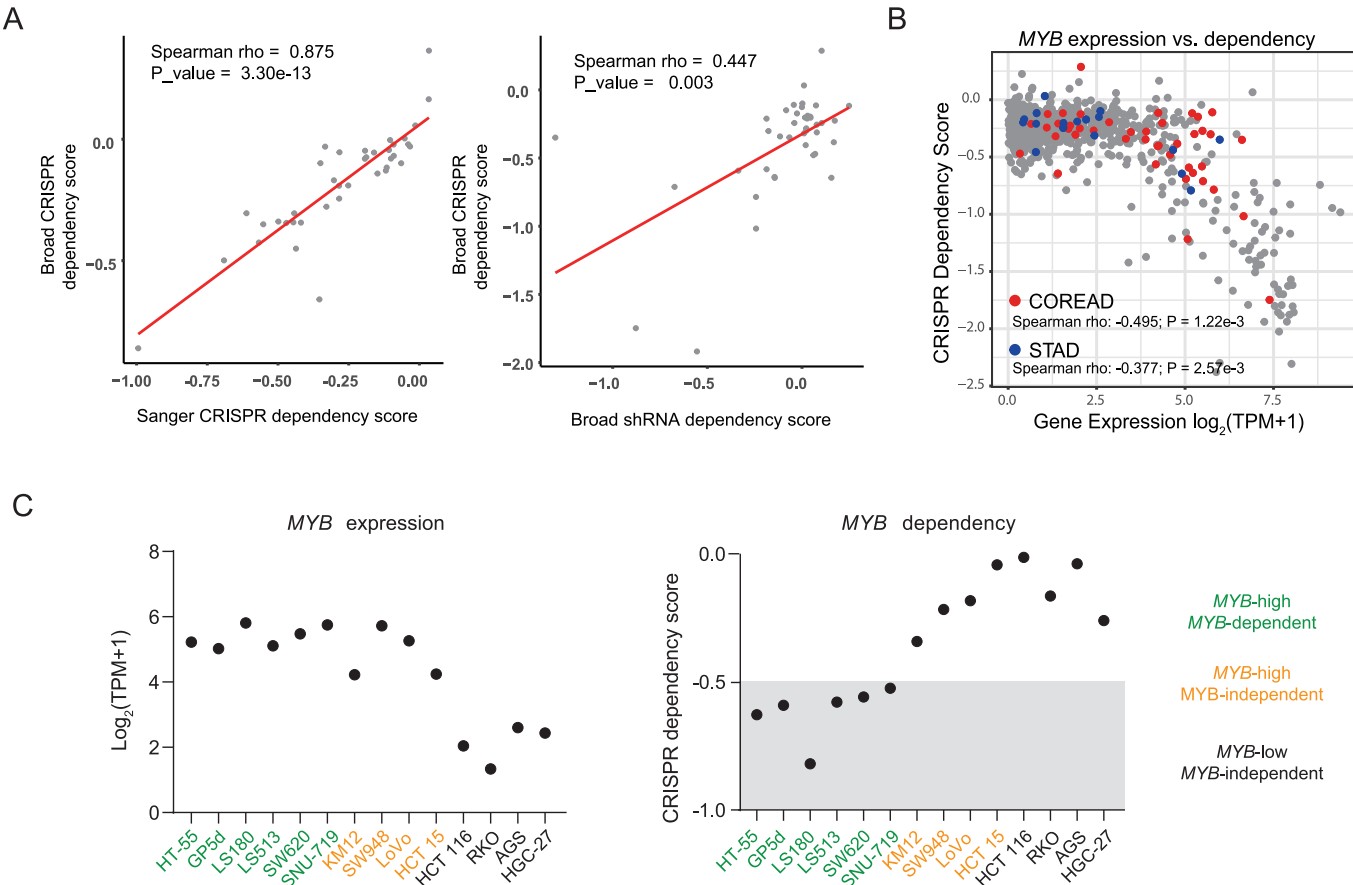

**Figure EV2. Functional significance of *MYB* in gastrointestinal adenocarcinoma.**

(**A**) Correlation between *MYB* dependency scores from DepMap and results from the Sanger Institute CRISPR screen (left) or the Broad Institute shRNA screen (right), calculated using Spearman's rank correlation coefficient based on the data from CCLE. (**B**) Correlation between *MYB* expression and dependency scores across all cancer cell lines in DepMap, calculated using Spearman's rank correlation coefficient based on the data from CCLE. The results of COREAD and STAD cell lines were highlighted in red and blue, respectively. (**C**) The expression levels and dependency scores of *MYB* in the indicated cell lines based on DepMap. Source data are available online for this figure.

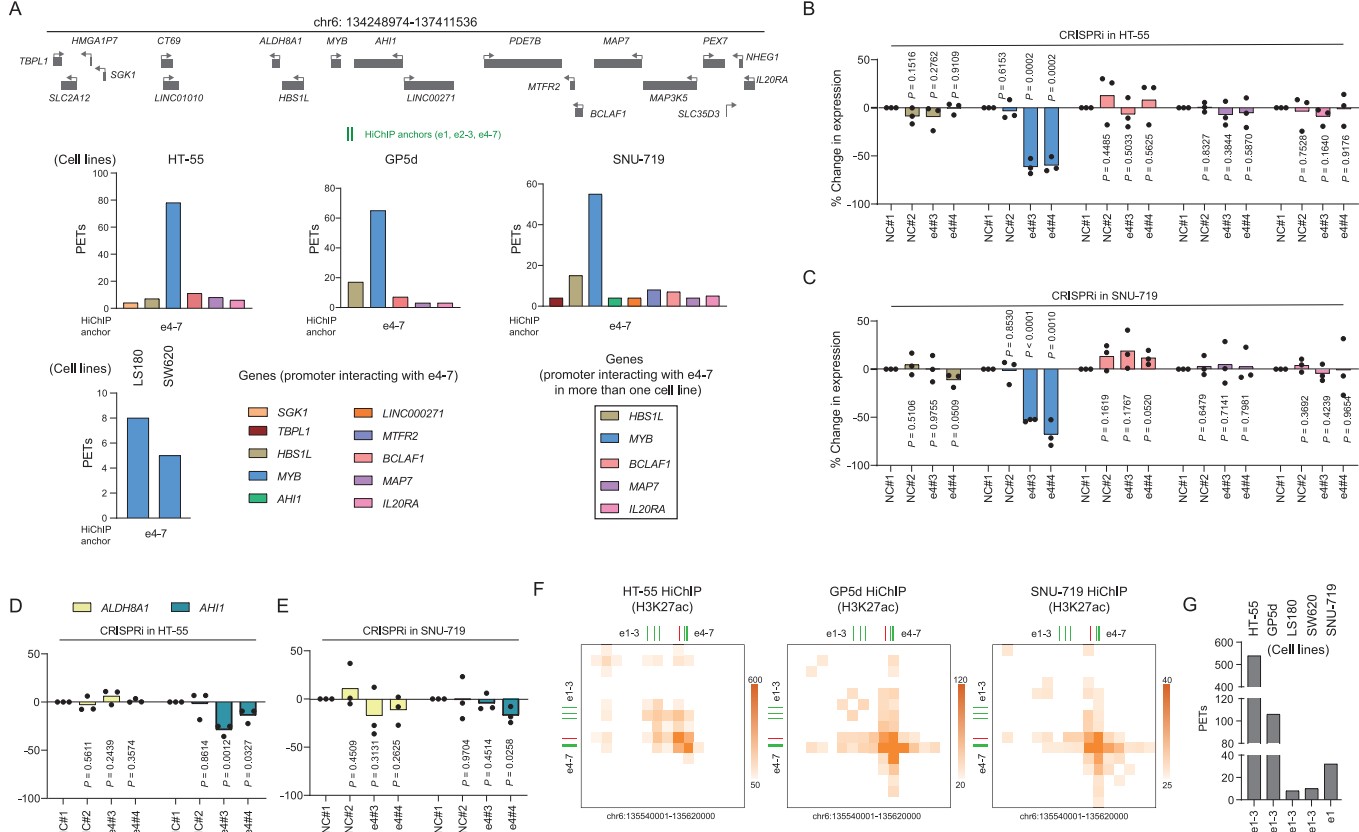

**Figure EV3. The enhancer e4 is the predominant element of the *MYB*-SE.**

(**A**) The number of paired-end tags (PETs) that connecting e4–7 anchor to the promoters of candidate genes in five cell lines, based on H3K27ac HiChIP data. (**B, C**) The expression of candidate genes following CRISPRi targeting e4 in HT-55 (**B**) and SNU-719 (**C**) cells. The genes were annotated with the same color as (**A**). All values are mean ± S.D. The *P*-value was determined by two-sided Student's *t*-test. (**D, E**) The expression of genes adjacent to *MYB* following CRISPRi targeting e4 in HT-55 (**D**) and SNU-719 (**E**) cells. The genes were annotated with the same color as (**A**). All values are mean ± S.D. The *P*-value was determined by two-sided Student's *t*-test. (**F**) The contact heatmap based on H3K27ac HiChIP data revealed that e4 interacted with other candidate enhancers within the *MYB*-SE in HT-55, GP5d, and SNU-719 cells. (**G**) The number of PETs that connecting e4–7 anchor to e1–3 or e1 anchor based on H3K27ac HiChIP data in five cell lines. Source data are available online for this figure.

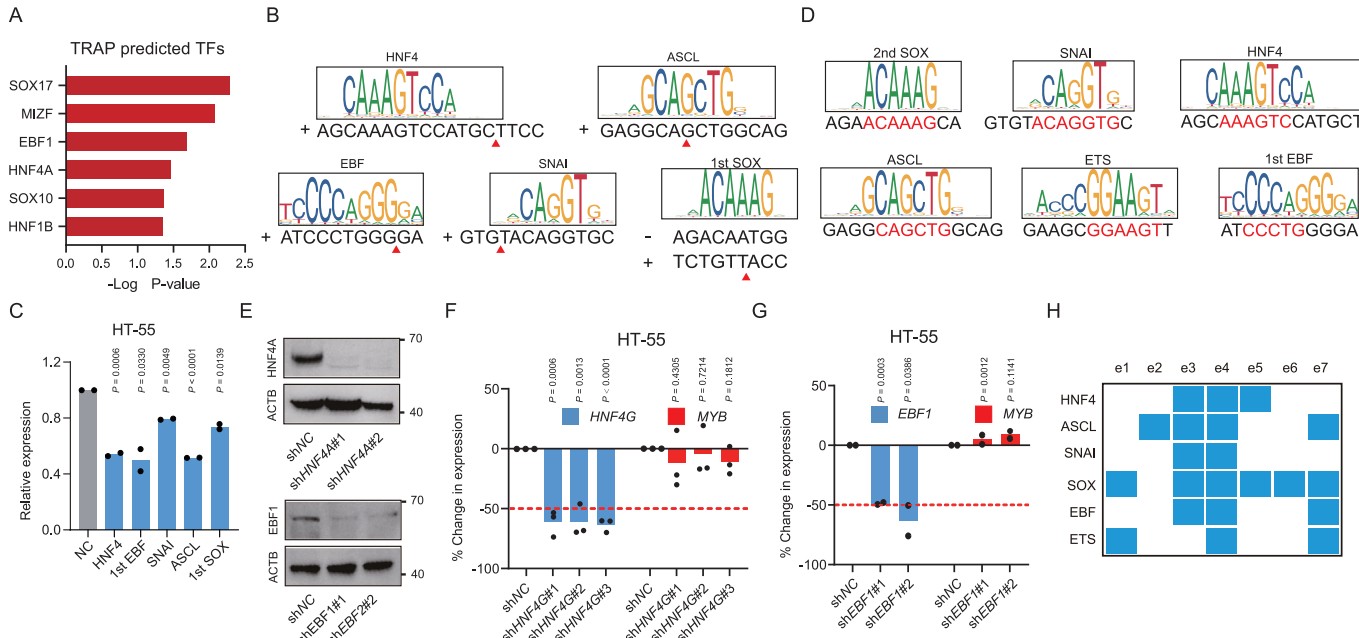

**Figure EV4. Candidate transcription factors regulating *MYB* expression through interactions with e4.**

(A) Transcription factors predicted to bind to e4 based on TRAP motif analysis; the *P*-value was determined by Benjamini-Hochberg multiple comparison correction method. (B) Demonstration of the motifs to be disrupted by CRISPR-Cas9. Red triangles indicated the cutting sites. (C) The expression of *MYB* in HT-55 cells following CRISPR-Cas9 based disruption of the identified motifs. N = 2. All values are mean ± S.D. The *P*-value was determined by two-sided Student's *t*-test. (D) Demonstration of the motifs to be deletion by site-specific mutagenesis for the Luciferase assay. Red sequences indicated the targets of deletion. (E) Western blotting showing a decrease in HNF4A and EBF1 protein levels upon *HNF4A* or *EBF1* knockdown, respectively. (F) The expression of *HNF4G* and *MYB* upon *HNF4G* knockdown. NC, non-targeted control. N = 3. All values are mean ± S.D. The *P*-value was determined by two-sided Student's *t*-test. (G) The expression of *EBF1* and *MYB* upon *EBF1* knockdown. NC, non-targeted control. N = 3. All values are mean ± S.D. The *P*-value was determined by two-sided Student's *t*-test. (H) Presence of the candidate functional motifs in the e1–e7 enhancers. Source data are available online for this figure.

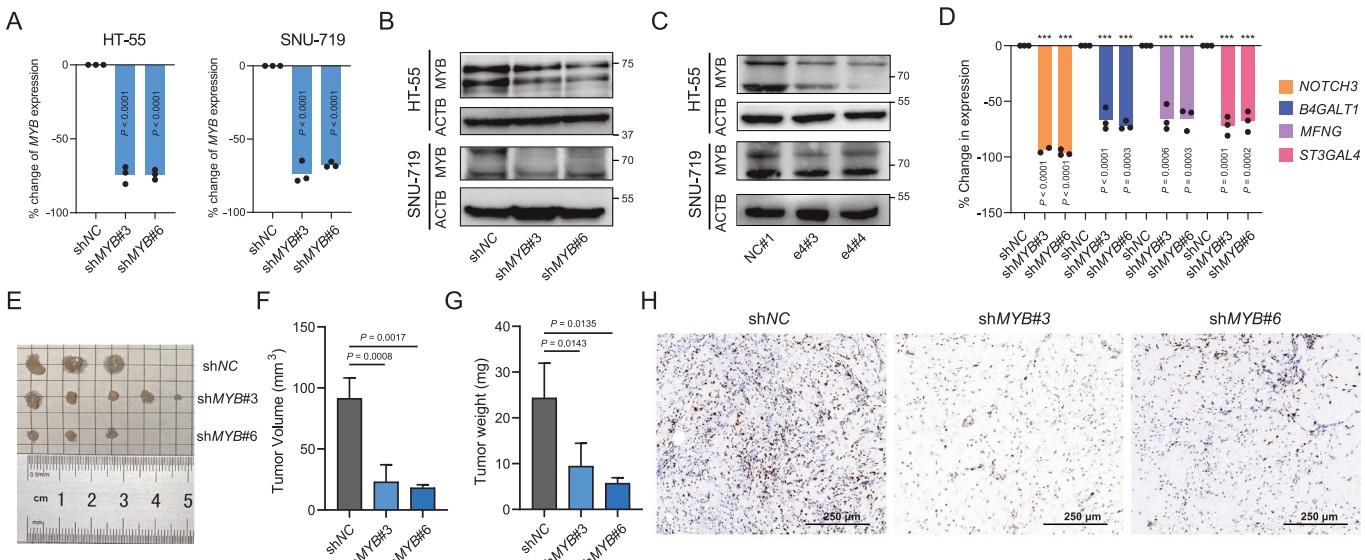

**Figure EV5. Repression of *MYB* expression in vivo inhibited the development of gastrointestinal adenocarcinoma.**

(A) The expression of *MYB* upon *MYB* knockdown in HT-55 and SNU-719 cells. N = 3. All values are mean ± S.D. The *P*-value was determined by two-sided Student's *t*-test. (B, C) Western blotting showing the decreased MYB protein upon *MYB* knockdown (B) or CRSPRi targeting e4 (C) in HT-55 and SNU-719 cells. The ACTB protein served as a loading control. (D) RT-qPCR analysis demonstrated the alterations in Notch signaling-related genes following *MYB* knockdown. N = 3. All values are mean ± S.D. The *P*-value was determined by two-sided Student's *t*-test. (E–G) The growth of the xenografts derived from HT-55 cells was significantly inhibited following *MYB* knockdown. N = 3–5. All values are mean ± S.D. The *P*-value was determined by two-sided Student's *t*-test. (H) Immunohistochemical staining for Ki-67 in xenograft samples following *MYB* knockdown in HT-55 cells. Scale bar = 250 µm. Source data are available online for this figure.

