## [Peer Review File · Molecular Systems Biology]

Systemic genome-epigenome analysis captures a lineage-specific super-enhancer for MYB in gastrointestinal cancer

Fuyuan Li, Shangzi Wang, Lian Chen, Ning Jiang, Xingdong Chen, and Jin Li

Corresponding author(s): Jin Li (li_jin_lifescience@fudan.edu.cn) , Xingdong Chen (xingdongchen@fudan.edu.cn)

Review Timeline:

Submission Date:	2nd Dec 24
Editorial Decision:	14th Jan 25
Revision Received:	2nd Mar 25
Editorial Decision:	24th Mar 25
Revision Received:	26th Mar 25
Accepted:	31st Mar 25

Editor: Jingyi Hou

Transaction Report:

14th Jan 2025

Manuscript Number: MSB-2024-12782

Title: Systemic genome-epigenome analysis captures a lineage-specific super-enhancer for MYB in gastrointestinal cancer

Author: Fuyuan Li

Shangzi Wang

Lian Chen

Ning Jiang

Xingdong Chen

Jin Li

Dear Jin,

Thank you for submitting your work to Molecular Systems Biology. We have now heard back from the three reviewers who agreed to evaluate your manuscript. As you will see from the reports below, the reviewers find the study interesting. They raise, however, a series of concerns, which we would ask you to address in a major revision.

The reviewers' recommendations are relatively clear, so there is no need to reiterate the points listed below. All the issues raised by the reviewers need to be satisfactorily addressed. As you may already know, our editorial policy allows in principle a single round of major revision, and it is therefore essential to provide responses to the reviewers' comments that are as complete as possible. Please feel free to contact me in case you would like to discuss in further detail any of the issues raised by the reviewers.

On a more editorial level, we would ask you to address the following issues:

- Please provide a .docx formatted version of the manuscript text (including legends for main figures, EV figures and tables). Please make sure that the changes are highlighted to be clearly visible.
- Please provide individual production quality figure files as .eps, .tif, .jpg (one file per figure).
- Please provide a .docx formatted letter INCLUDING the reviewers' reports and your detailed point-by-point responses to their comments. As part of the EMBO Press transparent editorial process, the point-by-point response is part of the Review Process File (RPF), which will be published alongside your paper.
- Please note that all corresponding authors are required to supply an ORCID ID for their name upon submission of a revised manuscript.
- We replaced Supplementary Information with Expanded View (EV) Figures and Tables that are collapsible/expandable online (see examples in <http://msb.embopress.org/content/11/6/812>). A maximum of 5 EV Figures can be typeset. EV Figures should be cited as 'Figure EV1, Figure EV2' etc... in the text and their respective legends should be included in the main text after the legends of regular figures.

Additional Tables/Datasets should be labeled and referred to as Table EV1, Dataset EV1, etc. Legends have to be provided in a separate tab in case of .xls files. Alternatively, the legend can be supplied as a separate text file (README) and zipped together with the Table/Dataset file.

For the figures and tables that you do NOT wish to display as Expanded View figures, they should be bundled together with their legends in a single PDF file called *Appendix*, which should start with a short Table of Content. Each legend should be below the corresponding Figure/Table in the Appendix. Appendix figures and tables should be referred to in the main text as: "Appendix Figure S1, Appendix Figure S2, Appendix Table S1" etc. See detailed instructions regarding expanded view here: <https://www.embopress.org/page/journal/17444292/authorguide#expandedview>.

- Before submitting your revision, primary datasets (and computer code, where appropriate) produced in this study need to be deposited in an appropriate public database (see <http://msb.embopress.org/authorguide-dataavailability> <https://www.embopress.org/page/journal/17444292/authorguide#dataavailability>).

The accession numbers and database should be listed in a formal "Data Availability" section (placed after Materials & Method) that follows the model below (see also <https://www.embopress.org/page/journal/17444292/authorguide#dataavailability>). Please note that the Data Availability Section is restricted to new primary data that are part of this study.

Data availability

- RNA-Seq data: Gene Expression Omnibus GSE46843 (<https://www.ncbi.nlm.nih.gov/geo/query/acc.cgi?acc=GSE46843>)

- [data type]: [name of the resource] [accession number/identifier/doi] ([URL or identifiers.org/DATABASE:ACCESSION])

-At EMBO Press we ask authors to provide source data for the main figures. Our source data coordinator will contact you to discuss which figure panels we would need source data for and will also provide you with helpful tips on how to upload and organize the files.

- Our journal encourages inclusion of *data citations in the reference list* to directly cite datasets that were re-used and obtained from public databases. Data citations in the article text are distinct from normal bibliographical citations and should directly link to the database records from which the data can be accessed. In the main text, data citations are formatted as follows: "Data ref: Smith et al, 2001". In the Reference list, data citations must be labeled with "[DATASET]". A data reference must provide the database name, accession number/identifiers and a resolvable link to the landing page from which the data can be accessed at the end of the reference. Further instructions are available at .

- We updated our journal's competing interests policy in January 2022 and request authors to consider both actual and perceived competing interests. Please review the policy <https://www.embopress.org/competing-interests> and update your competing interests if necessary.

Please use the heading "Disclosure statement and competing interests".

- All Materials and Methods need to be described in the main text using our 'Structured Methods' format. According to this format, the Methods section includes a Reagents and Tools Table (listing key reagents, experimental models, software and relevant equipment and including their sources and relevant identifiers) followed by a Methods and Protocols section describing the methods, ideally using a step-by-step protocol format. The aim is to facilitate adoption of the methodologies across labs. Please download and fill our Reagents and Tools Table template (.docx), which you can find in our author guidelines: <https://www.embopress.org/page/journal/17444292/authorguide#structuredmethods>.

- Regarding data quantification:

Please ensure to specify the name of the statistical test used to generate error bars and P values, the number (n) of independent experiments (please specify technical or biological replicates) underlying each data point and the test used to calculate p-values in each figure legend. Discussion of statistical methodology can be reported in the materials and methods section, but figure legends should contain a basic description of n, P and the test applied.

Graphs must include a description of the bars and the error bars (s.d., s.e.m.).

- Please provide a "standfirst text" summarizing the study in one or two sentences (approximately 250 characters, including space), three to four "bullet points" highlighting the main findings and a "synopsis image" (550px width and 400-600 px height, PNG format) to highlight the paper on our homepage.

Here are a couple of examples:

<https://www.embopress.org/doi/10.15252/msb.20199356>

<https://www.embopress.org/doi/10.15252/msb.20209475>

<https://www.embopress.org/doi/10.15252/msb.209495>

When you resubmit your manuscript, please download our CHECKLIST (<https://www.embopress.org/pb-assets/embo-site/EMBO%20Press%20Author%20Checklist-1642513524327.xlsx>) and include the completed form in your submission.

Please note that the Author Checklist will be published alongside the paper as part of the transparent process (<https://www.embopress.org/page/journal/17444292/authorguide#transparentprocess>).

If you feel you can satisfactorily deal with these points and those listed by the referees, you may wish to submit a revised version of your manuscript. Please attach a covering letter giving details of the way in which you have handled each of the points raised by the referees. A revised manuscript will be once again subject to review and you probably understand that we can give you no guarantee at this stage that the eventual outcome will be favorable.

I look forward to receiving the revised manuscript soon.

Kind regards,

Jingyi

Jingyi Hou, PhD
Senior Editor
Molecular Systems Biology

We realize that it is difficult to revise to a specific deadline. In the interest of protecting the conceptual advance provided by the work, we recommend a revision within 3 months (14th Apr 2025). Please discuss the revision progress ahead of this time with the editor if you require more time to complete the revisions. Use the link below to submit your revision:

IMPORTANT: When you send your revision, we will require the following items:

1. the manuscript text in LaTeX, RTF or MS Word format
2. a letter with a detailed description of the changes made in response to the referees. Please specify clearly the exact places in the text (pages and paragraphs) where each change has been made in response to each specific comment given
3. three to four 'bullet points' highlighting the main findings of your study
4. a short 'blurb' text summarizing in two sentences the study (max. 250 characters)
5. a 'thumbnail image' (550px width and max 400px height, Illustrator, PowerPoint or jpeg format), which can be used as 'visual title' for the synopsis section of your paper.
6. Please include an author contributions statement after the Acknowledgements section (see <https://www.embopress.org/page/journal/17444292/authorguide>)
7. Please complete the CHECKLIST available at (<https://bit.ly/EMBOPressAuthorChecklist>). Please note that the Author Checklist will be published alongside the paper as part of the transparent process (<https://www.embopress.org/page/journal/17444292/authorguide#transparentprocess>).
8. When assembling figures, please refer to our figure preparation guideline in order to ensure proper formatting and readability in print as well as on screen:

See also figure legend guidelines: <https://www.embopress.org/page/journal/17444292/authorguide#figureformat>

9. Please note that corresponding authors are required to supply an ORCID ID for their name upon submission of a revised manuscript (EMBO Press signed a joint statement to encourage ORCID adoption). (<https://www.embopress.org/page/journal/17444292/authorguide#editorialprocess>)
Currently, our records indicate that the ORCID for your account is 0000-0002-7957-1476.

Link Not Available

11. Include a Reagents and Tools Table as part of the Methods section, which can be downloaded from our author guidelines (<https://www.embopress.org/page/journal/17444292/authorguide#structuredmethods>)

*** PLEASE NOTE *** As part of the EMBO Press transparent editorial process initiative (see our Editorial at <https://dx.doi.org/10.1038/msb.2010.72>), Molecular Systems Biology publishes online a Review Process File with each accepted manuscripts. This file will be published in conjunction with your paper and will include the anonymous referee reports, your point-by-point response and all pertinent correspondence relating to the manuscript. If you do NOT want this File to be published, please inform the editorial office at msb@embo.org within 14 days upon receipt of the present letter.

Reviewer #1:

This manuscript by Li et al. presents a comprehensive investigation of a superenhancer (SE) downstream of the MYB gene, which seems to be specific for gastrointestinal adenocarcinoma. The authors integrate published H3K27ac ChIP-seq data from

various adenocarcinoma primary tumors and cell lines to identify the SE, which is supported by 3D interaction, MYB upregulation in adenocarcinoma, and occurrence in a recurrently duplicated genomic region. The constituent enhancers within the SE are then dissected in detail using a combination of luciferase assays, CRISPRi & CRISPRa, and comparison to published data, revealing that enhancer e4 is the predominant contributor to the SE's activity. The authors further identify the major transcription factors (TFs) controlling the activity of e4. Again, analyses of published datasets are combined with not only motif searches but also very detailed downstream confirmatory analysis including CRISPR to disrupt TF binding motifs. Further validation was done by TF ChIP, TF knockdown/inhibition combined with luciferase assays. Finally, the authors demonstrate that MYB knockdown and CRISPRi of e4 inhibit cell growth in vitro and in mouse xenografts. Overall, this very detailed dissection and validation of the MYB superenhancer will be valuable for the community. However, the clarity, flow and English language could be improved in some places. Some specific points:

- In the main text related to Fig 1A it is written "7897 and 4113 SEs in cancer cell lines and tumor samples". These high numbers are misleading because it seems that SE regions appearing in several cell lines are accounted for more than once. I think it would be better if the number of the union of SEs would be stated in a way that an SE region that is present in several cell lines/tumors is only counted once.
- In the second paragraph of the first results section the authors go quickly from describing 11 SE-oncogenes to picking MYB as it has the recurrent amplicon covering the noncoding region near MYB. Have the authors also checked for amplicons surrounding the other 10 SEs? It would be good if the authors could mark the location of their identified SE next to MYB in Fig 1F.
- Please indicate in Fig 1F which cell lines are shown in the amplification panel. Along those lines, since the other cell lines seem to have the same SE (based on Suppl Fig. 2C), I wonder if for the other cell lines there is simply no amplification data available or are there also cell lines that have the same SE but no amplification is observed? If the latter, does the amplification correlate with the signal of H3K27ac in Suppl Fig. 2C?
- Along those lines: Suppl Fig 2I shows no correlation between gene copy number and expression of MYB. Is there a correlation between MYB-SE amplification and expression of MYB?
- Fig 2: To better compare the CTCF peaks and motifs in A with the HiChIP data in B it would maybe be better to plot them on top of each other so they can be directly compared (i.e. by restricting the region in A to the same as in B). The extended range shown in A upstream and downstream of the region shown in B is not needed. Main text description of Fig 2A says MYB-SE resided in the same adjacent insulated neighborhood. I think this needs better explanation as either it is in the same or it is in the adjacent neighborhood.
- Fig 2B / Suppl Fig 3B: the authors compare MYB-high/-dependent cell lines to MYB-low/-independent. How do MYB-high/-independent cell lines behave? From Suppl. Fig 3C it gets clear that these also have high H3K27ac levels, so I would assume HiChIP would also show an interaction here? If so, how do the authors then explain the MYB-independence?
- It needs to be better/earlier described what the data from DepMap is / what MYB dependency means, especially for non experts. The authors actually describe it in a bit more detail a bit later in the paper when they describe Suppl Fig. 4A, but since there is already a large result section related to MYB dependency before, it would be good to describe it in more detail already before when the data is first used.
- When the authors describe the cooperative effect of the constituent enhancers do they wish to state a synergistic effect, meaning multiple enhancers working together to produce a greater effect than the sum of their individual effects? Or rather simply a cooperative effect?
- Since MYB seems to bind e4,6,7 and regulates e4, why was MYB not identified in the TFBS analysis presented in Fig 4A?

Typos:

- Intro: major cancer types should be type
- Intro: SEs -> use marked by rather than bound by H3K27ac
- First paragraph: "SEs assigned genes" I think assigned genes can be removed
- End of first results paragraph: remove "for"
- Beginning of second paragraph: should be "with their adjacent SE"
- Exhibited an even greater dependent on the MYB gene should be "dependence"
- STARR-seq comparison: compare should be compared
- When describing their CRISPRi approach "the an" should be "an"
- Suppl Fig 4F first panel should be GP5d?
- Suppl Fig 4G y axis labels: I assume it should be fold change rather than % change?

Reviewer #2:

The authors identified a super-enhancer for the MYB oncogene in gastric cancer, which also happens to be amplified in a subset of gastric cancer. They also identified potential transcription factors that are responsible for the activity of the super-enhancer. This is an interesting study that revealed an epigenetic mechanism activating MYB in gastric cancer. The experiments were well designed with appropriate controls included. The thorough dissection of the e4 enhancer is a plus. That being said, there are some parts in the manuscript that need further clarification, which will enhance the rigor of the work.

- 1) Figure 1B needs more clarification. How many SE are overlapped between cell line and tumors? How are the "adjacent"

genes defined? Also, how "oncogenes" are annotated?

2) Figure 1E, which cell lines are the HiChIP data derived from? It seems that the Supplementary Table 2 should be mentioned here.

3) It is not clear what "insulated neighborhoods" are defined in Figure 2A based on the CTCF motif orientations. To me, this panel is confusing and adds little information to the manuscript, especially given that HiChIP loops are clearly showing interactions at the MYB locus.

4) Figure S2E needs more clarification/annotation. The mouse/human conserved region, corresponding to the human SE, should be highlighted. It is not clear if it is the activity of the conserved region that is repressed in mouse or maybe the DNA sequence of the SE region is not conserved in mouse.

5) Can the authors check the numbers mentioned in "publicly available ATAC-seq datasets of tumor samples from 81 patients with primary COREAD and 41 patients with primary STAD from TCGA". The number of patients look higher than reported. Maybe the authors refer to the number of "samples" instead of "patients" (one patient/donor may have multiple samples)?

6) Fig 3F suggests a joint effect from the enhancers, instead of a "synergistic" effect, which should be corrected in the text.

7) I would recommend modify the model presented in figure 6K. Given the fact that most, if not all, of the tested cell lines don't have 'duplication' of the SE region and they still have high expression of MYB, I would recommend emphasize the function of the SE itself and remove the 'duplication' part in the model.

Minor:

There are quite a few typos and grammar errors in the manuscript, which need to be corrected. Also, "cancer oncogene" should be corrected to "oncogene".

Reviewer #3:

Initial statement

The work carried out by Li and colleagues assesses the involvement of MYB in the development of gastrointestinal adenocarcinomas and explores the role of a nearby super enhancer (MYB-SE) in its upregulation. MYB has been previously shown to be involved in the progression of colorectal cancers, however the mechanisms behind its upregulation are not fully understood. In this manuscript a novel mechanism, based on a non-coding structural variant involving the MYB-SE is reported and characterized.

General comments

1. The manuscript would greatly benefit from a general deep grammar revision. There are several linguistic mistakes that undermine the importance of the work presented in this manuscript and, at times, make it difficult for the reader to follow the experimental work presented.

2. In different moments of the manuscript the authors either oversimplify the current knowledge available or exaggerate the importance of their findings. For example, in the first paragraph of the introduction it is stated that "It has been validated that the variations of non-coding region in human genome have significant impacts on health¹⁻⁴. Therefore, it is reasonable to hypothesize that the non-coding variants also play an important role in the expression program of the oncogenes as well as the progress of cancer.", when in fact, non-coding variants have already been shown to affect oncogene expression and cancer progression. The authors actually refer to some of these works, however the phrasing used suggests this will be the time that non-coding variants are studied in the context of cancer. The manuscript would improve if the authors avoid such oversimplifications/ exaggerations.

3. There are several instances throughout the manuscript where the necessary references are needed. Here are some examples:

"By nature, the transcriptional regulation by the non-coding regions is highly lineage-specific"

"Dysregulation of transcriptional programs can be mediated by epigenetic alterations targeting the non-coding regulatory elements like enhancers and super-enhancers (SEs)."

"It has been demonstrated that the recurrent alterations of SEs play an important role in tumorigenesis and malignant phenotypes by interacting with the corresponding promoters via the three-dimensional (3D) structure of genome."

"The overexpression of MYB is widely observed in cancer cells and the oncogenic activity of MYB has been confirmed in leukemia, breast cancer, prostate cancer, et al."

Major points

1. Regarding the CRISPRi and CRISPRa experiments additional controls are need to confidently draw conclusions. For example, targeting a region that has not regulatory features located at an approximate similar distance or in the case of the CRISPRa experiments the targeting of other parts of the SE (e1, e2 or e3).

Minor points

1. Regarding figure 1A, what were the H3K27ac signal cutoff used to determine an enhancer was a super enhancer?
2. The authors mentioned that they observed a recurrent amplicon in the noncoding region near the MYB gene. In figure 1F it is represented several amplicons present in that area of the genome. In other occasions they refer to the existence of a duplication that overlaps with the existing SE. It is not clear whether a duplication of the same sequence is present across different cancer cells, or different duplications in that area have been reported. It would be important to clarify this as this is one of the observations that drives the investigation of the MYB-SE.
3. Although the genome positions are included in the presented tracks, it would be very useful to include a scale.
4. "Although the expression of AHI1 surrounding MYB was modestly affected, no interaction with e4 was identified by the analysis of H3K27ac HiChIP data (Supplementary Fig. 5D-E), suggesting this is not a direct gene target of e4." Different models have been proposed regarding enhancer regulation mechanism that do not include loop extrusion. The statement above suggests this is the only model. A rephrasing of this sentence is necessary.
5. The authors identify HNF4, EBF, ASCL, SOX and SNAI motifs in e4. It would be interesting to include the information whether any of these motifs are present in other parts of the SE (e1-3 and e5-7).
6. Regarding the knockdown of EBF1, please include a western blot that shows the protein levels upon knockdown of the gene.

We appreciate reviewers' valuable comments and suggestions on our manuscript. Below, we provide detailed responses to address each point, accompanied by the corresponding revisions in the manuscript. We believe the reviewers' suggestions have significantly improved our manuscript.

Reviewer 1 Comments for the Author...

This manuscript by Li et al. presents a comprehensive investigation of a superenhancer (SE) downstream of the MYB gene, which seems to be specific for gastrointestinal adenocarcinoma. The authors integrate published H3K27ac ChIP-seq data from various adenocarcinoma primary tumors and cell lines to identify the SE, which is supported by 3D interaction, MYB upregulation in adenocarcinoma, and occurrence in a recurrently duplicated genomic region. The constituent enhancers within the SE are then dissected in detail using a combination of luciferase assays, CRISPRi & CRISPRa, and comparison to published data, revealing that enhancer e4 is the predominant contributor to the SE's activity. The authors further identify the major transcription factors (TFs) controlling the activity of e4. Again, analyses of published datasets are combined with not only motif searches but also very detailed downstream confirmatory analysis including CRISPR to disrupt TF binding motifs. Further validation was done by TF ChIP, TF knockdown/inhibition combined with luciferase assays. Finally, the authors demonstrate that MYB knockdown and CRISPRi of e4 inhibit cell growth in vitro and in mouse xenografts. Overall, this very detailed dissection and validation of the MYB superenhancer will be valuable for the community.

We appreciate the reviewer's positive comments about our manuscript.

However, the clarity, flow and English language could be improved in some places. Some specific points:

- In the main text related to Fig 1A it is written "7897 and 4113 SEs in cancer cell lines and tumor samples". These high numbers are misleading because it seems that SE regions appearing in several cell lines are accounted for more than once. I think it would be better if the number of the union of SEs would be stated in a way that an SE region that is present in several cell lines/tumors is only counted once.

We agree with the reviewer and apologize for the confusion. Indeed, cancer cell lines have 3433 unique SEs and tumor samples have 2137 unique SEs. We have added the following sentences in lines 77-78 of the revised manuscript: "*We identified 3433 unique SEs in cancer cell lines and 2137 unique SEs in tumor samples, respectively (Fig. 1A).*"

- In the second paragraph of the first results section the authors go quickly from describing 11 SE- oncogenes to picking MYB as it has the recurrent amplicon covering the noncoding region near MYB. Have the authors also checked for amplicons surrounding the other 10 SEs?

We have checked if all 11 SEs are located within recurrent duplication genomic regions. Among them, only the SEs targeting *KLF5* and *MYB* fall within these regions. In addition, the duplication of the *KLF5*-SE has been reported previously (Zhang *et al*, 2018). We have added the following sentences in lines 93-96 of the revised manuscript: “Among these 11 SEs, only the SEs associated with *KLF5* and *MYB* were located within the genomic regions showing recurrent duplication, as observed in a high-resolution whole-genome sequencing dataset from the Pan-Cancer Atlas of Whole Genomes (PCAWG) Project (Consortium, 2020). The *KLF5*-SE duplication has been reported previously (Zhang *et al.*, 2018).”

- It would be good if the authors could mark the location of their identified SE next to MYB in Fig 1F.

We are grateful of the suggestions from the reviewer. We have now modified the Fig. 1F and marked the location of *MYB*-SE.

Fig. 1F: The *MYB* amplicons across gastrointestinal adenocarcinoma samples from PCAWG project and merged enhancer profile from cell lines at the *MYB*-associated locus in gastrointestinal adenocarcinoma.

- Please indicate in Fig 1F which cell lines are shown in the amplification panel. Along those lines, since the other cell lines seem to have the same SE (based on Suppl Fig. 2C), I wonder if for the other cell lines there is simply no amplification data available or are there also cell lines that have the same SE but no amplification is observed? If the latter, does the

amplification correlate with the signal of H3K27ac in Suppl Fig. 2C?

We appreciate that the reviewer points out this important issue. For Fig. 1F, the duplication data is derived from WGS data of patient samples in PCAWG project, in which the H3K27ac ChIP-seq data is not available. The H3K27ac signal track are derived from cell lines, which have no corresponding duplication data, either. Therefore, we cannot determine whether the H3K27ac signal of *MYB-SE* is associated with its amplification. To clarify this point, we have revised the manuscript and added the following sentences in lines 100-104: *“The information of duplication was derived from WGS data of tumor samples, which the H3K27ac ChIP-seq data is not available. The information of H3K27ac binding sites was derived from cell lines, which have no corresponding WGS data. Therefore, we cannot determine whether the H3K27ac signal of MYB-SE is associated with its amplification.”*

The detailed presentation of H3K27ac in individual cell line is shown in Supplemental Fig. 2C. To clarify this point, we have now explained it in figure legend of Supplemental Fig. 2C (Supplementary Fig. 2C is now referred to as Fig. EV1C): *“The cell line information provides a detailed description of the cell lines depicted in Fig. 1F.”*

- Along those lines: Suppl Fig 2I shows no correlation between gene copy number and expression of *MYB*. Is there a correlation between *MYB-SE* amplification and expression of *MYB*?

We agree with the reviewer that it is interesting to investigate the correlation between the copy number of *MYB-SE* and the expression of *MYB*. However, only one *MYB-SE* duplication sample of gastrointestinal adenocarcinoma has the data for both RNA-seq and WGS data in the PCAWG project, which makes it impossible to answer this question currently.

- Fig 2: To better compare the CTCF peaks and motifs in A with the HiChIP data in B it would maybe be better to plot them on top of each other so they can be directly compared (i.e. by restricting the region in A to the same as in B). The extended range shown in A upstream and downstream of the region shown in B is not needed. Main text description of Fig 2A says *MYB-SE* resided in the same adjacent insulated neighborhood. I think this needs better explanation as either it is in the same or it is in the adjacent neighborhood.

We apologize for any inconvenience caused by inappropriate images. Based on the reviewer's

suggestion, we have now modified Fig. 2 by adjusting the scale of panels A and B to the same region and marking the chromatin loop domain as following:

Fig.2A-B. A: The CTCF ChIP-seq tracks of multiple gastrointestinal cancer cell lines were presented. The red and green arrows represented the orientation of the CTCF motifs. A schematic of the chromosomal *MYB* loop domain is shown at the bottom; B. The chromatin loops based on H3K27ac HiChIP data connected the *MYB* promoter to multiple distal elements with high-confidence PETs in various gastrointestinal cancer cell lines.

We have also revised the manuscript by adding the following sentences lines 149-151: “Interestingly, the *MYB* promoter and the candidate *MYB-SE* resided within the same chromatin loop domain, defined by mutually convergent CTCFs (Fig. 2A), suggesting that the *MYB* promoter may interact with the candidate *MYB-SE* through chromatin looping.”

- Fig 2B / Suppl Fig 3B: the authors compare *MYB*-high/-dependent cell lines to *MYB*-low/-independent. How do *MYB*-high/-independent cell lines behave? From Suppl. Fig 3C it gets clear that these also have high H3K27ac levels, so I would assume HiChIP would also show an interaction here? If so, how do the authors then explain the *MYB*-independence?

We agree with the reviewer that the inconsistency between *MYB* expression and dependence is an interesting topic to investigate. We analyzed H3K27ac HiChIP data of HT-29 cells, a *MYB*-high/-independent cell line with high H3K27ac levels in *MYB-SE*. We did notice a strong interaction between the enhancer e4 and the *MYB* promoter (Figure R1). Therefore, in some cells with *MYB* high/-independent, the enhancer e4 can also interact with the *MYB* promoter.

Figure R1: Chromatin loops based on H3K27ac HiChIP data connected the *MYB* promoter to multiple distal elements with high-confidence PETs in the HT-29 cell line.

The independence on *MYB* in *MYB*-high cell lines can be explained by two reasons. 1. Due to the complexity of cancer cells, it is possible that some cell lines can survive with alternative mechanism when *MYB* expression is suppressed; 2. We noticed that the stitched enhancer at e4 does not form an SE in HT-29 cells, which may also contribute to the regulation of dependence on *MYB*. We have revised the manuscript by adding the following sentences in lines 155-161: “*The results of the HiChIP analysis indicated that the constituent enhancers of the candidate MYB-SE had highly confident pair-end-tags (false discovery rate (FDR) < 0.05) linking them to the MYB promoter in MYB-high/-dependent cell lines (Fig. 2B) and a MYB-high/-independent HT-29 cell line (Appendix Fig. S2A). In contrast, these pair-end-tags were absent in MYB-low/-independent cell lines (including HCT 116, RKO, AGS cell lines) and normal human primary colonic epithelial cells (Appendix Fig. S2B). The independence of HT-29 on MYB can be attributed to either potential alternative mechanism of survival or the absence of SE (Appendix Fig. S2A).*”

- It needs to be better/earlier described what the data from DepMap is / what *MYB* dependency means, especially for non experts. The authors actually describe it in a bit more detail a bit later in the paper when they describe Suppl Fig. 4A, but since there is already a large result section related to *MYB* dependency before, it would be good to describe it in more detail already before when the data is first used.

We agree with the reviewer and have included this information in lines 126-133 of the revised manuscript: “*To determine the functional significance of MYB in gastrointestinal adenocarcinoma, we analyzed the genome-wide CRISPR screen data released by DepMap (Tsherniak et al, 2017). The CRISPR dependency score was normalized such that, for each*

cell line, the median effect of a set of known non-essential genes (referred to as "negative controls") is 0, and the median effect of known essential genes (referred to as "positive controls") is -1 (Liu et al., 2020). As a result, a lower score indicates a higher likelihood that a gene is dependent in a given cell line, so a cutoff of -0.5 corresponds to half the effect size expected from knocking out genes that are essential for each cell line (Liu et al., 2020). Hence, if a cell line exhibit CRISPR dependency score < -0.5 for a given gene, the cell line is dependent on that gene."

- When the authors describe the cooperative effect of the constituent enhancers do they wish to state a synergistic effect, meaning multiple enhancers working together to produce a greater effect than the sum of their individual effects? Or rather simply a cooperative effect?

We apologize for the confusion. In fact, except for e4, the activity of the other candidate enhancers is very low, and their impact on *MYB* expression is minimal, as shown in Fig. 3B and Appendix Fig. S3D. Therefore, these enhancers exhibited a joint effect rather than a synergistic effect. We have revised the manuscript by adding the following sentences in lines 210-212: "*The luciferase assay demonstrated that the combination of e4 with other constituent enhancers within the MYB-SE exhibited higher activity than e4 alone (Fig. 3F), indicating a joint effect of these enhancers on transcriptional regulation.*"

- Since MYB seems to bind e4,6,7 and regulates e4, why was MYB not identified in the TFBS analysis presented in Fig 4A?

The reviewer indeed brought up an interesting topic. We used the non-redundant JASPAR CORE Position Frequency Matrix data for vertebrates to perform TFBS analysis. The MYB motif was actually identified by the analysis with the *P*-value as 0.005 (Figure R2). Specifically, the second MYB motif "ATGTCCACGTC" significantly overlapped with the SNAI motif, whose deletion reduced the activity of the enhancer e4 (Fig. 4B). However, the *P*-value threshold we used for the original analysis was 0.0001. So, the MYB motif was filtered out of the results. If we included the MYB motif to the results, 3000 more motifs were added and the false discovery rate may dramatically increase. Therefore, we believe that it is more suitable to present the analysis results with a *P*-value of 0.0001 in Fig. 4A.

Figure R2: The motif analysis of the enhancer e4 sequence. Bottom: The MYB motif predicted by FIMO in the enhancer e4. $P < 0.005$.

Typos:

- Intro: major cancer types should be type

We thank the reviewer for pointing out this writing error. We have corrected this word to “type” in lines 38-39 of the revised manuscript as per your suggestion: “*For instance, gastrointestinal adenocarcinoma is a major cancer type of the digestive system, ranking as the top cause of cancer-related deaths worldwide¹⁴.*”

- Intro: SEs -> use marked by rather than bound by H3K27ac

We thank the reviewer for pointing out this writing error. We have replaced “bound” with “marked” in lines 43-45 of the revised manuscript: “*In cancer research, SEs are described as clusters of enhancers in close proximity, marked by H3K27ac, transcription factors or mediator complex to drive the high expression of lineage-specific oncogenes (Hnisz et al, 201)*”.

- First paragraph: "SEs assigned genes" I think assigned genes can be removed

We appreciate the reviewer for this suggestion. We have removed the phrase "assigned genes" from the sentence in lines 79-81 of the revised manuscript: “*Cell lines and tumors shared 82 and 185 SEs, respectively, which are associated with 205 unique protein-coding genes (Fig. 1B and Table EV1), including 34 oncogenes (Fig. 1C).*”

- End of first results paragraph: remove "for"

We thank the reviewer for pointing out this writing error. We have removed “for” the sentence “*These findings suggested that the common SEs may play an...*” in lines 85-86 of the revised manuscript.

- Beginning of second paragraph: should be "with their adjacent SE"

We thank the reviewer for this comment. We have changed “Among the genes associated to common SEs” to “*With their adjacent SEs*” in line 87 of the revised manuscript according to the reviewer’s suggestion.

- Exhibited an even greater dependent on the MYB gene should be "dependence"

We appreciate the reviewer for pointing out this writing error. We have changed “dependent” to “dependence” in lines 136-138 of the revised manuscript: “*Importantly, the cancer cell lines with higher expression levels of MYB exhibited an even greater dependence on the MYB gene for many types of cancer.....*”

- STARR-seq comparison: compare should be compared

We appreciate the reviewer for pointing out this writing error. We have changed “compare” to “compared” in lines 176-177 of the revised manuscript: “*The results revealed that e4 exhibited stronger transcriptional activity compared to*”.

- When describing their CRISPRi approach "the an" should be "an"

We appreciate the reviewer for pointing out this writing error. We have changed “the an” to “an” in lines 184-186 of the revised manuscript: “*We performed an improved CRISPR-based interference (CRISPRi) approach.....*”.

- Suppl Fig 4F first panel should be GP5d?

We thank the reviewer for this observation. We have changed “GP2d” to “GP5d” in Appendix Fig.S3D. (Supplementary Fig. 4F is now referred to as Appendix Fig. S3D)

Appendix Fig. S3D: The effects of CRISPRi on the candidate enhancers regulating *MYB* expression in GP5d, CCK-81, HT-55, and SNU-719 cells. N = 3. All values are mean \pm S.D. The *P*-value was determined by Student's *t*-test. *: $P \leq 0.05$; ***: $P \leq 0.001$.

- Suppl Fig 4G y axis labels: I assume it should be fold change rather than % change?

We thank the reviewer for this observation. We have changed “% change of *MYB* expression” to “Fold change” in Supplementary Fig.4G. (Supplementary Fig. 4G is now referred to as Appendix Fig. S3E)

Appendix Fig. S3E: The effects of CRISPRa on the enhancer e4 regulating *MYB* expression in two *MYB*-low/-independent cell lines AGS and HGC-27. N = 2. All values are mean \pm S.D. The *P*-value was determined by Student's *t*-test. *: $P \leq 0.05$; **: $P \leq 0.01$; ***: $P \leq 0.001$.

Reviewer 2 Comments for the Author...

The authors identified a super-enhancer for the MYB oncogene in gastric cancer, which also happens to be amplified a subset of gastric cancer. They also identified potential transcription factors that are responsible for the activity of the super-enhancer. This is an interesting study that revealed an epigenetic mechanism activating MYB in gastric cancer. The experiments were well designed with appropriate controls included. The thorough dissection of the e4 enhancer is a plus. That being said, there are some parts in the manuscript that need further clarification, which will enhance the rigor of the work.

We are grateful of the positive comments from the reviewers.

1) Figure 1B needs more clarification. How many SE are overlapped between cell line and tumors? How are the "adjacent" genes defined? Also, how "oncogenes" are annotated?

We thank the reviewer for these questions. In the original manuscript, we have shown 32 overlapping SEs in both the cell lines and tumors in Supplementary Fig. 2B. In the revised manuscript, we have now listed the 27 protein-coding genes nominated by these 32 SEs and highlighted oncogenes in red (Supplementary Fig. 2B is now referred to as Fig. EV1B).

Figure EV1B: Venn diagram showing the shared SEs and their associated protein-genes between gastrointestinal cancer cell lines and tumor samples, with oncogenes highlighted in red.

We apologize for the use of the confusing word "adjacent" to describe the SEs associated genes. We used the genes annotated by ROSE as the associated genes of SEs, where the gene's promoter is the closest to the SE (in terms of linear distance). We have now revised this description in lines 79-81 of the revised manuscript: "*Cell lines and tumors shared 82 and 185 SEs, respectively, which are associated with 205 unique protein-coding genes (Fig. 1B and Table EV1), including 34 oncogenes (Fig. 1C).*"

For the annotation of oncogenes, we used three databases COSMIC, OncoKB and ONGene. This information has been described in the Appendix Material and Methods: "*Identification of*

oncogenes based on annotated information from COSMIC (<https://cancer.sanger.ac.uk/census>) (Sondka et al, 2018), OncoKB (<https://www.oncokb.org/cancerGenes>) (Chakravarty et al, 2017) and OGene (<https://ongene.bioinfo-minzhao.org/>) (Liu et al, 2017) databases.”

2) Figure 1E, which cell lines are the HiChIP data derived from? It seems that the Supplementary Table 2 should be mentioned here.

We apologize for the confusion and appreciate the reviewer’s suggestion. The HiChIP data in Fig. 1E was derived from HT-55 and SNU-719 cell lines. We have added this information in lines 87-89 of the revised manuscript: “With their closest SEs, 44 genes had physical interactions with SEs based on H3K27ac HiChIP analysis in HT55 and SNU-719 cells (Table EV2) ……”

3) It is not clear what "insulated neighborhoods" are defined in Figure 2A based on the CTCF motif orientations. To me, this panel is confusing and adds little information to the manuscript, especially given that HiChIP loops are clearly showing interactions at the MYB locus.

We apologize for any inconvenience caused by inappropriate images. We have now modified Fig. 2A and marking the chromatin loop domain as following:

Fig.2A: The CTCF ChIP-seq tracks of multiple gastrointestinal cancer cell lines were presented. The red and green arrows represented the orientation of the CTCF motifs; a schematic of the chromosomal MYB loop domain is shown at the bottom.

4) Figure S2E needs more clarification/annotation. The mouse/human conserved region, corresponding to the human SE, should be highlighted. It is not clear if it is the activity of the conserved region that is repressed in mouse or maybe the DNA sequence of the SE region is not conserved in mouse.

We agree with the reviewer that a detailed annotation of the conservation of MYB-SE in both human and mouse is worthwhile. Based on the reviewer’s suggestion, we converted the human MYB-SE gene coordinates to mouse coordinates (mm10) using the liftOver tool. We found that the most of the elements in MYB-SE is conserved between human and mouse

except the enhancer e6 (as shown in Fig. EV1F). We have now described this data in lines 114-116 of the revised manuscript: “We applied liftOver (Kent et al, 2002) to convert the human MYB-SE gene coordinates to mouse coordinates (mm10), and found that the MYB-SE is conserved between human and mouse except the enhancer e6 (Fig. EV1F)”.

(Supplementary Fig. 2F is now referred to as Fig. EV1F)

Fig. EV1F: ChIP-seq signal tracks of H3K27ac at the *Myb* and *Myc* loci in mouse CRC cell lines. The region of *MYB-SE* is conserved between humans and mice except the enhancer e6. However, the activity of human gastrointestinal adenocarcinoma cancer-specific super-enhancer of *MYB* (converted to mm10 genome) is repressed in this region in mouse CRC cell lines.

5) Can the authors check the numbers mentioned in "publicly available ATAC-seq datasets of tumor samples from 81 patients with primary COREAD and 41 patients with primary STAD from TCGA". The number of patients look higher than reported. Maybe the authors refer to the number of "samples" instead of "patients" (one patient/donor may have multiple samples)?

We appreciate that the reviewer points out an important mistake in our manuscript. We found that some patients indeed have multiple samples in ATAC-seq database released by TCGA. We have now corrected the number of donors in lines 219-221 of the revised manuscript: “To identify TFs that may bind to e4, we performed motif analysis using publicly available ATAC-seq datasets from tumor samples, including 81 primary COREAD samples (from 38 donors) and 41 primary STAD samples (from 21 donors) from TCGA (Corces et al., 2018).”

6) Fig 3F suggests a joint effect from the enhancers, instead of a "synergistic" effect, which should be corrected in the text.

The reviewer is right that the interactions between these enhancers are indeed a joint effect rather than a synergistic effect. We have now changed “synergistic” to “joint” in lines 211-212

in the revised manuscript: “.....indicating a joint effect of these enhancers within the *MYB-SE* on transcriptional regulation”.

7) I would recommend modify the model presented in figure 6K. Given the fact that most, if not all, of the tested cell lines don't have 'duplication' of the SE region and they still have high expression of MYB, I would recommend emphasize the function of the SE itself and remove the 'duplication' part in the model.

We fully agree with the reviewer that, in cell lines without *MYB-SE* amplification, *MYB-SE* can also promote high-level expression of *MYB*. We appreciate the reviewer’s suggestion and have made the revision to Fig. 6K as following:

Fig.6K: Schematic diagram: HNF4A and MYB bind to the enhancer e4 within *MYB-SE* to activate the expression of *MYB*. The overexpressed MYB protein directly upregulated its target genes and promoted the development of gastrointestinal adenocarcinoma.

Minor:

There are quite a few typos and grammar errors in the manuscript, which need to be corrected. Also, "cancer oncogene" should be corrected to "oncogene".

We thank the reviewer’s comment and have corrected the typos and grammar errors in the manuscript. We hope our manuscript has improved. And we have removed the word "cancer" from the sentence in lines 110-111 of the revised manuscript: “*In addition, though MYB is considered as an oncogene in AML with a potential super-enhancer (Pelish et al, 2015)*”

Reviewer 3 Comments for the Author...

Initial statement

The work carried out by Li and colleagues assesses the involvement of MYB in the development of gastrointestinal adenocarcinomas and explores the role of a nearby super enhancer (MYB-SE) in its upregulation. MYB has been previously shown to be involved in the progression of colorectal cancers, however the mechanisms behind its upregulation are not fully understood. In this manuscript a novel mechanism, based on a non-coding structural variant involving the MYB-SE is reported and characterized.

We appreciate the reviewer's positive comments regarding our manuscript.

General comments

1. The manuscript would greatly benefit from a general deep grammar revision. There are several linguistic mistakes that undermine the importance of the work presented in this manuscript and, at times, make it difficult for the reader to follow the experimental work presented.

We appreciate the reviewer's comment and apologize for the errors in the manuscript. We have now corrected the typos and grammar errors in the manuscript.

2. In different moments of the manuscript the authors either oversimplify the current knowledge available or exaggerate the importance of their findings. For example, in the first paragraph of the introduction it is stated that "It has been validated that the variations of non-coding region in human genome have significant impacts on health¹⁻⁴. Therefore, it is reasonable to hypothesize that the non-coding variants also play an important role in the expression program of the oncogenes as well as the progress of cancer.", when in fact, non-coding variants have already been shown to affect oncogene expression and cancer progression. The authors actually refer to some of these works, however the phrasing used suggests this will be the time that non-coding variants are studied in the context of cancer. The manuscript would improve if the authors avoid such oversimplifications/ exaggerations.

We appreciate the reviewer for this comment. We have now revised the manuscript and added the following sentences in lines 29-35: *"It has been validated that the variations of non-coding regions in the human genome have significant impacts on health (Groschel et al,*

2014; Kataoka et al, 2016; Mansour et al, 2014; Northcott et al, 2014; Peifer et al, 2015). For example, a partial loss of the 3' UTR of the PD-L1 gene is associated with the upregulation of PD-L1 expression in various cancer types, including T cell leukemia and B cell lymphoma (Kataoka et al., 2016). Somatic mutations in the TAL1 enhancer can introduce de novo MYB binding motifs, leading to the formation of a super-enhancer that drives aberrant TAL1 expression (Mansour et al., 2014).”

3. There are several instances throughout the manuscript where the necessary references are needed.

Here are some examples:

"By nature, the transcriptional regulation by the non-coding regions is highly lineage-specific"

"Dysregulation of transcriptional programs can be mediated by epigenetic alterations targeting the non-coding regulatory elements like enhancers and super-enhancers (SEs)."

"It has been demonstrated that the recurrent alterations of SEs play an important role in tumorigenesis and malignant phenotypes by interacting with the corresponding promoters via the three-dimensional (3D) structure of genome."

"The overexpression of MYB is widely observed in cancer cells and the oncogenic activity of MYB has been confirmed in leukemia, breast cancer, prostate cancer, et al."

We appreciate the reviewer’s comment and we have now added the references in revised manuscript. Such as the sentence in lines 35-36: “*Transcriptional regulation by non-coding regions is highly lineage-specific (Madani Tonekaboni et al, 2019).*”, the sentence in lines 41-43: “*Dysregulation of transcriptional programs can be driven by epigenetic alterations targeting non-coding regulatory elements such as enhancers and super-enhancers (SEs) (Leeman-Neill et al, 2023; Zhang & Meyerson, 2020; Zhou & Parsons, 2023).*”, the sentence in lines 45-48: “*It has been demonstrated that the recurrent alterations of SEs play an important role in tumorigenesis and malignant phenotypes by interacting with the corresponding promoters via the three-dimensional (3D) structure of genome (Liu et al, 2021; Xing et al, 2019; Zhang et al, 2018; Zhang et al, 2016).*” and the sentence in lines 56-59: “*The overexpression of MYB is widely observed in cancer cells and its oncogenic activity has been confirmed in leukemia (Pattabiraman & Gonda, 2013), breast cancer (Quintana et al., 2011), prostate cancer (Acharya et al, 2023) and other cancers (Ramsay & Gonda, 2008).*”

Major points

1. Regarding the CRISPRi and CRISPRa experiments additional controls are need to confidently draw conclusions. For example, targeting a region that has not regulatory features located at an approximate similar distance or in the case of the CRISPRa experiments the targeting of other parts of the SE (e1, e2 or e3).

Based on the reviewer's suggestion, we have now performed additional controls experiments to expand our functional validation. The results showed that CRISPRi of the control sgRNA upstream of e1 and the control sgRNA between e3 and e4 (sgCtrl_2) had minimal effects on the expression levels of *MYB* (Fig. R3A). Also, activation of e1 or e2 by CRISPRa can't activate the expression of *MYB* (Fig. R3B). These new results have been included in Fig. 3E and Figure Appendix Fig. S3E.

Figure R3: The expression of *MYB* upon CRISPRi-mediated repression of e4. Two separate sgRNAs, as sg-e4#3 and sg-e4#4, were used to target e4. NC, non-targeted control. sgCtrl_1 targeted region upstream of e1; sgCtrl_2 targeted region between e3 and e4. N = 3. All values are mean \pm S.D. The *P* value was determined by Student's *t* test. *: $P \leq 0.05$; **: $P \leq 0.01$; ***: $P \leq 0.001$.

Minor points

1. Regarding figure 1A, what were the H3K27ac signal cutoff used to determine an enhancer was a super enhancer?

We thank the reviewer for this comment. According to the description of ROSE algorithm to identify super enhancers (Hnisz *et al*, 2013), the H3K27ac signal cutoff should depend on the data of samples or cell lines, instead of a general cutoff. As described by Hnisz and colleagues in their research paper (in Supplemental information): "Briefly, this algorithm stitches constituent enhancers together if they are within a certain distance and ranks the enhancers by their input-subtracted signal of H3K27ac. It then separates super enhancers

from typical enhancers by identifying an inflection point of H3K27ac signal versus enhancer rank.” (Hnisz et al., 2013). In other words, a cutoff determined by plotting the distribution of H3K27ac ChIP-seq intensity values (both stitched and single enhancers) and designating enhancer regions to the right of the point at which the slope of the plot was 1 as super-enhancers, and the remaining enhancer regions with an inflection < 1 were defined as typical-enhancers (Fig. R4). Therefore, the cutoff for identifying SEs using the H3K27ac signal is different across different cell lines and cells.

Figure R4: Defining super-enhancer. Step 1: enhancer loci are defined by calling peaks on H3K27ac ChIP-seq data (The promoter is excluded); Step 2: enhancers within 12.5 kb (or a custom distance) of each other are combined into stitched enhancer regions. Step 3: ChIP-seq signal values for both stitched enhancer regions and enhancers without partners (also called single enhancers) are ranked along the X-axis on the basis of the H3K27ac enrichment plotted on the Y-axis. Super-enhancers are defined as regions to the right of the inflection point of the resulting curve (highlighted in pink).

2. The authors mentioned that they observed a recurrent amplicon in the noncoding region near the MYB gene. In figure 1F it is represented several amplicons present in that area of the genome. In other occasions they refer to the existence of a duplication that overlaps with the existing SE. It is not clear whether a duplication of the same sequence is present across different cancer cells, or different duplications in that area have been reported. It would be important to clarify this as this is one of the observations that drives the investigation of the MYB-SE.

We agree with the reviewer that it would be important to clarify whether a duplication of the MYB-SE sequence is present across different cancer cells. Focusing on focal duplications (< 1

Mb in length) across 37 cancer types in PCAWG project, we found that *MYB*-SE duplication was also present in other cancers, such as BRCA, OV, HCC and PRAD. In addition, the duplication for genomic region of *MYB*-SE has been reported in some of the T-ALL samples (Lahortiga *et al.*, 2007; O'Neil *et al.*, 2007). However, the *MYB*-SE region in ALL is not functional for *MYB* expression, as the enhancers in ALL for *MYB* are located 116 kb away from the *MYB*-SE (Li *et al.*, 2021; Stadhouders *et al.*, 2014).

Previous studies have shown that duplications tend to cluster together in cancers bearing biological similarities, and recurrent duplications (also known as duplication hotspots) exhibit high cancer-type specificity (Song *et al.*, 2024). In the current study, we found *MYB*-SE is frequently duplicated in GI cancer in comparison to other cancer types (7/198, 2/168, 1/109, 1/289, 1/146 for GI, BRCA, OV, HCC and PRAD, respectively) (Fig. R5). We have now added this information in lines of 99-100 of the revised manuscript: “*Although duplications containing MYB-SE exist in many cancer types, MYB-SE duplication occurs more frequently in gastrointestinal adenocarcinoma (Fig. EVID).*” and added the follow sentence in lines of 308-312 in our Discussion section of the revised manuscript: “*We also found MYB-SE is frequently duplicated in gastrointestinal adenocarcinoma in comparison to other cancer types. Although the duplication for genomic region of MYB-SE has been reported in some of the T-ALL samples (Lahortiga et al., 2007; O'Neil et al., 2007), the MYB-SE region in ALL is not functional for MYB expression, as the enhancers in ALL for MYB are located 116 kb away from the MYB-SE (Li et al., 2021; Stadhouders et al., 2014).*”

Figure R5: Focal duplications harboring *MYB*-SE across 37 cancer types in PCAWG project.

3. Although the genome positions are included in the presented tracks, it would be very useful to include a scale.

We thank the reviewer for this suggestion. To clarify, we have labeled the scale of the signal on the right side of the Y-axis of each track plot.

4. "Although the expression of *AHI1* surrounding *MYB* was modestly affected, no interaction

with e4 was identified by the analysis of H3K27ac HiChIP data (Supplementary Fig. 5D-E), suggesting this is not a direct gene target of e4." Different models have been proposed regarding enhancer regulation mechanism that do not include loop extrusion. The statement above suggests this is the only model. A rephrasing of this sentence is necessary.

We thank the reviewer for this suggestion. We have now rephrased this sentence in lines 203-207 of the revised manuscript as: *"Although the expression of AHI, a gene located near MYB, was modestly affected, no direct interaction with e4 was identified by the analysis of H3K27ac HiChIP data (Fig. EV3D-E). This suggests that e4 does not directly target AHI promoter via a chromatin loop extrusion mechanism; however, other enhancer regulation models, such as phase separation (Hnisz et al, 2017; Tang et al, 2022), could still contribute to the gene's regulation."*

5. The authors identify HNF4, EBF, ASCL, SOX and SNAI motifs in e4. It would be interesting to include the information whether any of these motifs are present in other parts of the SE (e1-3 and e5-7).

We agree with the reviewer that it would be interesting to identify determine the HNF4, EBF, ASCL, SOX and SNAI motifs in e4 are present in the other enhancers of MYB-SE. We performed motif analysis on the other enhancers of MYB-SE, and found some of the candidate functional motifs in e4 were also present in other enhancers (Fig. R6). For example, the SOX motif was also present in enhancers e1, e3, e5, e6, and e7, and the HNF4 motif was also present in e3 and e5. This information has been included in Fig. EV4H in the revised manuscript and described in line 240-242 of the revised manuscript: *"Additionally, we found several candidate functional transcription factor motifs in other enhancers, suggesting that these transcription factors may bind to multiple enhancers to collaboratively regulate the activity of the MYB-SE (Fig. EV4H)."*

Figure R6: Presence of the candidate functional motifs in the e1–e7 enhancers

6. Regarding the knockdown of EBF1, please include a western blot that shows the protein levels upon knockdown of the gene.

Based on the reviewer's suggestion, we have now applied western blot to measure the expression levels of EBF1 after *EBF1*-KD (Fig. R7). This information has been included in Fig. EV4E.

Figure R7: Western blotting showing a decrease in EBF1 protein levels upon *EBF1* knockdown in HT-55 cells.

References:

- Barretina J, Caponigro G, Stransky N, Venkatesan K, Margolin AA, Kim S, Wilson CJ, Lehar J, Kryukov GV, Sonkin D *et al* (2012) The Cancer Cell Line Encyclopedia enables predictive modelling of anticancer drug sensitivity. *Nature* 483: 603-607
- Ghandi M, Huang FW, Jane-Valbuena J, Kryukov GV, Lo CC, McDonald ER, 3rd, Barretina J, Gelfand ET, Bielski CM, Li H *et al* (2019) Next-generation characterization of the Cancer Cell Line Encyclopedia. *Nature* 569: 503-508
- Hnisz D, Abraham BJ, Lee TI, Lau A, Saint-Andre V, Sigova AA, Hoke HA, Young RA (2013) Super-enhancers in the control of cell identity and disease. *Cell* 155: 934-947
- Lahortiga I, De Keersmaecker K, Van Vlierberghe P, Graux C, Cauwelier B, Lambert F, Mentens N, Beverloo HB, Pieters R, Speleman F *et al* (2007) Duplication of the MYB oncogene in T cell acute lymphoblastic leukemia. *Nat Genet* 39: 593-595
- Li M, Jiang P, Cheng K, Zhang Z, Lan S, Li X, Zhao L, Wang Y, Wang X, Chen J *et al* (2021) Regulation of MYB by distal enhancer elements in human myeloid leukemia. *Cell Death Dis* 12: 223
- Loven J, Hoke HA, Lin CY, Lau A, Orlando DA, Vakoc CR, Bradner JE, Lee TI, Young RA (2013) Selective inhibition of tumor oncogenes by disruption of super-enhancers. *Cell* 153: 320-334
- O'Neil J, Tchinda J, Gutierrez A, Moreau L, Maser RS, Wong KK, Li W, McKenna K, Liu XS, Feng B *et al* (2007) Alu elements mediate MYB gene tandem duplication in human T-ALL. *J Exp Med* 204: 3059-3066
- Song Y, Li F, Wang S, Wang Y, Lai C, Chen L, Jiang N, Li J, Chen X, Bailey SD *et al* (2024) Chromatin interaction maps identify oncogenic targets of enhancer duplications in cancer. *Genome Res* 34: 1514-1527
- Stadhouders R, Aktuna S, Thongjuea S, Aghajani-refah A, Pourfarzad F, van Ijcken W, Lenhard B, Rooks H, Best S, Menzel S *et al* (2014) HBS1L-MYB intergenic variants modulate fetal hemoglobin via long-range MYB enhancers. *J Clin Invest* 124: 1699-1710
- Zhang X, Choi PS, Francis JM, Gao GF, Campbell JD, Ramachandran A, Mitsuishi Y, Ha G, Shih J, Vazquez F *et al* (2018) Somatic Superenhancer Duplications and Hotspot Mutations Lead to Oncogenic Activation of the KLF5 Transcription Factor. *Cancer Discov* 8: 108-125

24th Mar 2025

Manuscript Number: MSB-2024-12782R

Title: Systemic genome-epigenome analysis captures a lineage-specific super-enhancer for MYB in gastrointestinal cancer

Author: Fuyuan Li

Shangzi Wang

Lian Chen

Ning Jiang

Xingdong Chen

Jin Li

Dear Jin,

Thank you for sending us your revised manuscript. We have now heard back from the three reviewers who were asked to evaluate your revised study. As you will see below, the reviewers are satisfied with the performed revisions and support publication. Before we can formally accept the manuscript for publication, we would ask you to address some remaining editorial-level issues listed below.

1. Please address the remaining minor issues raised by Reviewer #1.
2. Please remove all the figures from the manuscript file. Figure legends should remain in manuscript file. Figures should be uploaded as individual files, with one figure per file. Please remove other versions to avoid confusion.
3. "Financial support statement" should be included in the "Acknowledgements". Ensure the funding information entered in the submission matches those in the manuscript file. Specifically, the grant number SKLGE-2118 is currently missing in the online system.
4. Please remove the "Author contribution" section from the manuscript file.
5. Expanded Tables should be uploaded as individual Excel files with the corresponding legends included in each Excel file. The nomenclature should be Table EV1-EV6.
6. Source data:
 - Source data files for Fig. 5B-F, 5H, 6F and 6I are missing- this needs to be addressed or clarified in the source data checklist,
 - Source data files should be organized in a scheme one figure/folder and then uploaded as .zip files. E.g. all the Source data files for figure 1 need to be saved in a single folder and this needs to be zipped and then uploaded as "SD figure 1.zip" file.
 - For EV and/or appendix figures, all source data should be zipped together.
 - Completed SD checklist should be uploaded as Related Manuscript File.
7. Data availability: please ensure the datasets will be made publicly available upon the acceptance of the manuscript.
8. I have slightly modified the synopsis text (see attached). Please let me know if it is fine as is or if you would like to introduce further modifications.
9. Author checklist: In the "Journal submitted to" box, the journal name should be entered instead of the corresponding author's name.
10. Appendix:
 - The title page should contain "Appendix for manuscript title" and a Table of Content with the page numbers for the listed items.
 - All the Appendix figures need to be compiled into one PDF, with legends placed below corresponding figures.
11. Please add callout(s) for Table EV6.
12. Section order should be corrected to Title page - Abstract & Keywords - Introduction - Results - Discussion - Methods - DataAvailability - Acknowledgements - Disclosure and CompetingInterests Statement - References - Figure Legends - Table(s) - Expanded View Figure Legends.
13. Please address the following issues related to legends:
 - Please note that the exact p values are not provided in the legends of figures 3B, C, E, F; 4B,C, D, E, F, G, H; 5G, 6A, B, C, D, E, G, H, J; EV1 H, I; EV3 B, C, D,E; EV4 C, F, G; EV5 A, D, F, G.
 - Please indicate the statistical test used for data analysis in the legends of figures 1D, 5D, EV4 A

- Please note that in figures 5G there is a mismatch between the annotated p values in the figure legend and the annotated p values in the figure file that should be corrected.
- Please note that information related to n is missing in the legends of figures EV11.
- Please note that the box plots need to be defined in terms of minima, maxima, centre, bounds of box and whiskers, and percentile in the legends of figures EV1 H, I.

When you resubmit your manuscript, please download our CHECKLIST (<https://bit.ly/EMBOPressAuthorChecklist>) and include the completed form in your submission. *Please note* that the Author Checklist will be published alongside the paper as part of the transparent process (<https://www.embopress.org/page/journal/17444292/authorguide#transparentprocess>)

Click on the link below to submit your revised paper.

Thank you for submitting this interesting paper to Molecular Systems Biology.

Sincerely,
Jingyi

Jingyi Hou, PhD
Senior Editor
Molecular Systems Biology

*** PLEASE NOTE *** As part of the EMBO Press transparent editorial process initiative (see our Editorial at <https://dx.doi.org/10.1038/msb.2010.72> , Molecular Systems Biology will publish online a Review Process File to accompany accepted manuscripts. When preparing your letter of response, please be aware that in the event of acceptance, your cover letter/point-by-point document will be included as part of this File, which will be available to the scientific community. More information about this initiative is available in our Instructions to Authors. If you have any questions about this initiative, please contact the editorial office (msb@embo.org).

Reviewer #1:

All my concerns/suggestions have been addressed and I recommend publication.

I would advise the authors to check the grammar/typos carefully again.

E.g.

l. 41: remove "is" from "is remains"

l. 55: remove "containing"

l. 105: should be ", for which"

l. 106: "information of H3K27ac binding sites was derived" should be "information on H3K27ac was derived"

l. 115: "presented" should be "present"

l. 120: maybe better to write "the DNA sequence underlying the MYB-SE is conserved"

l. 140: should be "correlated"

l. 152: "in" missing

l. 224: should be "regulate"

l. 225: remove "the" or "this"

l. 296: should be "knockdown of"

Reviewer #2:

The authors have largely addressed my previous concerns by providing additional clarification and analyses. In my view, the revised manuscript is acceptable to MSB.

Reviewer #3:

The author have addressed my concerns, the additonal experiments and controls have significantly improved the manuscript. In my opinion the article is ready to be published.

All editorial and formatting issues were resolved by the authors.

31st Mar 2025

Manuscript number: MSB-2024-12782RR

Title: Systemic genome-epigenome analysis captures a lineage-specific super-enhancer for MYB in gastrointestinal cancer

Dear Jin,

Thank you again for sending us your revised manuscript. We are now satisfied with the modifications made and I am pleased to inform you that your paper has been accepted for publication.

Yours sincerely,
Jingyi

Jingyi Hou, PhD
Senior Editor
Molecular Systems Biology
